# Evaluating the Inductive Abilities of Large Language Models: Why Chain-of-Thought Reasoning Sometimes Hurts More Than Helps

**Haibo Jin**
School of Information Sciences
University of Illinois at Urbana-Champaign
Champaign, IL 61820
`haibo@illinois.edu`

**Peiyan Zhang**
Computer Science and Engineering
HKUST
Clear Water Bay, Kowloon, Hong Kong
`pzhangao@connect.ust.hk`

**Man Luo**
Research Scientist, Intel Labs
Santa Clara, CA 95054
`luoman.cs@gmail.com`

**Haohan Wang**[*]
School of Information Sciences
University of Illinois Urbana-Champaign
Champaign, IL 61820
`haohanw@illinois.edu`

## Abstract

Large Language Models (LLMs) have shown remarkable progress across domains, yet their ability to perform inductive reasoning—inferring latent rules from sparse examples—remains limited. It is often assumed that chain-of-thought (CoT) prompting, as used in Large Reasoning Models (LRMs), enhances such reasoning. We investigate this assumption with creating four controlled, diagnostic game-based tasks—chess, Texas Hold'em, dice games, and blackjack—with hidden human-defined rules. We find that CoT reasoning can degrade inductive performance, with LRMs often underperforming their non-reasoning counterparts.

To explain this, we present a theoretical framework that reveals how reasoning steps can amplify error through three failure modes: incorrect sub-task decomposition, incorrect sub-task solving, and incorrect final answer summarization. Based on our theoretical and empirical analysis, we introduce structured interventions that adapt CoT generation according to our identified failure types. These interventions improve inductive accuracy without retraining. Our findings suggest that effective (CoT) reasoning depends not only on taking more steps but also on ensuring those steps are well-structured.

## 1 Introduction

Inductive reasoning—inferring latent rules from sparse examples—is a key capability underlying generalization. While Large Language Models (LLMs) [1, 2] have made significant advances, they continue to struggle in tasks that require structured inference under uncertainty [3]. These models often rely on surface-level pattern recognition and static prompt formats [4], making them brittle when confronted with novel or structurally complex problems [5].

To address this limitation, recent models incorporate explicit reasoning mechanisms—particularly chain-of-thought (CoT) prompting [6]—to enable multi-step inference. These Large Reasoning Models (LRMs) [7, 8] have shown improved performance in coding, logical inference, and scientific

---

[*]Corresponding Author

39th Conference on Neural Information Processing Systems (NeurIPS 2025).

tasks [9, 10], and CoT has been widely adopted under the assumption that structured reasoning enhances inductive performance [11, 12, 13]. However, emerging evidence complicates this assumption: Wu et al. [14] observe that longer reasoning traces can degrade accuracy, suggesting a non-monotonic relationship between reasoning depth and performance.

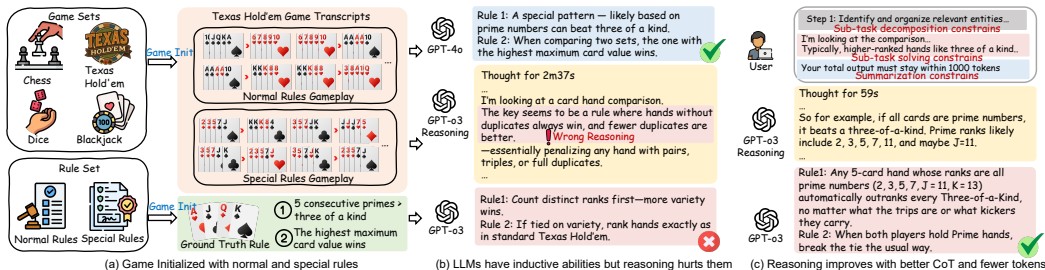

Figure 1: Examples illustrating inductive reasoning on gameplay transcripts. (a) Games begin with both Normal and hidden Special Rules, requiring models to infer latent constraints from observed plays. (b) LLMs can induce rules like card legality and win conditions without explicit guidance, but LRMs such as GPT-o3 may underperform due to misaligned or noisy reasoning. (c) Reasoning improves when guided at the decomposition, solving, and summarization stages.

In this work, we investigate the inductive performance of LLMs and LRMs (Fig. 1). We introduce a set of controlled diagnostic game-based tasks to isolate inductive reasoning behavior in LLMs. In each task, models are presented with a short transcript of gameplay—without access to the underlying rules—and must infer the latent constraints governing legal moves and outcomes. Surprisingly, we find that LRMs often underperform non-reasoning LLMs in these settings, suggesting that CoT reasoning may introduce noise rather than clarity. We also develop a theoretical framework that explains this degradation.

Our work address the following questions: **RQ1:** How well do LLMs perform on inductive reasoning tasks, and has this improved with recent models? (Section 3) **RQ2:** Why does reasoning sometimes fail—or even hurt—inductive performance? (Section 4) **RQ3:** How can we guide reasoning to enhance inductive accuracy without model retraining? (Section 5)

Along answer these RQs, we have made the following contributions.

- We construct four controlled diagnostic game-based tasks with hidden human-defined rules to evaluate the inductive reasoning abilities of eight leading LLMs, including both non-reasoning models and LRMs.
- We provide a theoretical and empirical analysis identifying three failure modes that explain degraded reasoning: incorrect sub-task decomposition, incorrect sub-task solving, and incorrect final answer summarization.
- We propose an error-guided intervention method that adapts CoT generation based on predicted failure types, yielding consistent improvements in inductive accuracy without retraining.

## 2 Related Work

**Chain-of-Thought Reasoning.** Large language models can perform complex tasks more effectively when guided through intermediate reasoning steps, a process known as chain-of-thought (CoT) prompting [6]. Much recent work has aimed to strengthen CoT-based reasoning through better decomposition, exploration, or decision-making strategies. Zhou et al. [11] introduce least-to-most prompting, which breaks problems into simpler subproblems and solves them sequentially to support easy-to-hard generalization. Yao et al. [12] propose Tree of Thoughts, enabling multi-path exploration with backtracking and self-evaluation. Meng et al. [13] introduce a Divide-and-Conquer strategy that improves performance by segmenting input into reasoning chunks, and Zhang et al. [15] evaluate when such strategies yield consistent gains. While these methods aim to enhance reasoning capabilities, they largely assume that reasoning depth is beneficial and do not investigate when CoT may fail.

**Mechanisms Behind CoT Reasoning.** Several studies seek to formalize the mechanisms underlying CoT prompting. Feng et al. [16] use circuit complexity to show that constant-size models can

simulate deeper computation via CoT derivations. Li et al. [17] explain CoT's benefits in decoder-only Transformers via sequential computation theory, and Cui et al. [18] show that Coherent CoT improves error correction over stepwise prompting but increases sensitivity to intermediate-step failures. Ton et al. [19] apply information theory to quantify stepwise information gain, and Li et al. [20] analyze fast vs. slow thinking during post-training. Ye et al. [21] investigate CoT in a controlled math setting, while Wu et al. [14] demonstrate that performance may peak and then decline with longer reasoning traces, hinting at an optimal CoT length. However, while these works identify performance degradation empirically, they stop short of explaining why it arises. Our work builds on these insights by offering a formal theory of reasoning failure.

**Inductive reasoning.** Inductive reasoning [22] is a core component of human intelligence, enabling rule generalization from examples without prior knowledge. As LLMs reach human-level performance, many benchmarks have been developed to assess this ability. Banatt et al. [23] propose WILT, a logic induction benchmark where models infer Boolean rules through trial-and-error queries. Li et al. [24] design MIRAGE, a suite of synthetic analogy tasks that test pattern-based generalization. Ma et al. [25] introduce KoR-Bench, which emphasizes rule application across logic puzzles, ciphers, and counterfactual reasoning. Xiao et al. [26] present LogicVista, focusing on visual-spatial rule inference via figure completion tasks. Xu et al. [27] develop LLM-Script, where models must induce latent functional patterns from input–output examples. Yan et al. [28] introduce MIR-Bench, the first benchmark for many-shot inductive reasoning, where models must induce latent functions from diverse input–output exemplars presented in long-context settings.

**Key Differences.** Existing benchmarks are typically static, using simplified symbolic or visual inputs to test whether models can apply hidden rules to new examples. In contrast, our game-based setup asks models to infer the rule itself from minimal gameplay context, without a separate testing phase. This more closely mirrors human induction, which relies on observing outcomes and forming hypotheses rather than answering predefined queries.

## 3 Inductive Reasoning Evaluation via Gameplay Tasks

To investigate how well LLMs perform in inductive reasoning tasks (RQ1), we introduce four different types of controlled diagnostic game-based tasks designed to challenge and measure their inductive ability to infer hidden rules from gameplay transcripts. These games span a diverse range of domains, including chess, Texas Hold'em, dice games, and blackjack. Each game consists of two rule components: a **normal rule** component (abbreviated as "**NR**") that mirrors standard gameplay conventions, and a **special rule** component (abbreviated as "**SR**") that introduces manually designed hidden constraints. Together, these components form a composite logic that governs legality, winning conditions, or valid moves. Crucially, these rules are not revealed to the models; instead, the models are presented with a few legal gameplay examples and are required to induce the underlying rules through observation.

### 3.1 Game Setup

**Models**. We evaluate eight language models, including four non-reasoning LLMs—GPT-4o (`gpt-4o-2024-08-06`) [1], DeepSeek-V3 (`deepseek-v3-2025-03-25`) [29], Qwen2.5-Max (`qwen-max-2025-01-25`) [30], and Grok-2 (`grok-2-1212`) [31]—as well as four LRMs: GPT-o3 (`o3-2025-04-16`), DeepSeek-R1 (`deepseek-r1-32b`) [7], QwQ (`QwQ-32B`) [32], and Grok-3-Mini (`grok-3-mini-beta`) [33]. We run DeepSeek-R1 and QwQ locally to enable collection of intermediate reasoning traces for deeper analysis in RQ2, while the remaining models are accessed via their respective APIs.

**Implementation Details.** To evaluate each model's inductive performance across rule types, we frame the task as a semantic alignment problem—determining whether the model's induced rule captures the same logic as the ground-truth. This formulation is supported by recent work showing that large language models can reliably perform semantic evaluation and alignment tasks [34]. We therefore use GPT-4o (`gpt-4o-2024-08-06`)[1] as a reference judge. For each case, GPT-4o is queried three times to independently assess whether the induced rule is consistent with the ground-truth, and a majority vote is used to finalize the decision. Rule-level accuracy is computed as the proportion of cases judged as consistent. Representative examples are provided in Appendix A.1, and the prompt for GPT-4o is provided in Appendix A.3.

## 3.2 Rule Design

We evaluate inductive reasoning in LLMs using four game-based tasks: chess, Texas Hold'em, dice games, and blackjack. Each task presents models with gameplay transcripts governed by two types of rules: normal (NRs) and special (SRs). The model must infer these rules purely from observation.

**Case Study: Chess.**  We use chess as a detailed example due to its well-defined rules and spatial structure. In each game instance, we define eight types of pieces, each assigned one rule: either from a pool of normal rules (NRs) or special rules (SRs). Normal rules are based on standard chess movements, such as moving one square in any direction or diagonally across the board. Special rules are designed to be legal but non-obvious, introducing hidden constraints that models must discover from context. For example, a piece may be assigned a rule like "move in a straight line any number of squares, then shift diagonally by one square." This rule differs subtly from standard movement and cannot be inferred from a single move.

Models are never given any rule descriptions—they are shown short gameplay transcripts (typically 10–12 moves) and asked to determine what rules govern the movements. We randomize the board size (ranging from 8 to 15), shuffle rule assignments across games, and ensure each piece appears at least three times per episode. We construct 225 distinct transcripts by enumerating all combinations of four NRs and four SRs across the six available options. This design provides a broad set of inductive scenarios with controlled structure and consistent constraints. Full rules and prompt formats are in Appendix B and D.1.

**Other Games.**  We include three additional games to cover a broader range of inductive reasoning scenarios, each characterized by distinct rule structures and abstraction types. Full rule specifications for all games are provided in Appendix B:

- **Texas Hold'em:** Tests symbolic ranking under hidden structure. While normal rules follow standard poker hand rankings, SRs redefine hand strength using patterns like "five consecutive prime numbers".
- **Dice Games:** Adapted from Sic Bo to assess reasoning in noisy environments. NRs involve numeric thresholds (e.g., small vs. large totals), while SRs introduce subtle structural overrides—e.g., "a triple of prime numbers beats all other hands".
- **Blackjack:** Focuses on threshold logic with rule exceptions. NRs include standard conditions like "busts lose" and "21 wins," while SRs add constraints such as "hands with three-card straight flushes win automatically".

## 3.3 Inductive Abilities Analysis

Fig. 2 shows rule-wise inductive accuracy across eight models in four games. Although designed to enhance multi-step reasoning, models with reasoning capabilities consistently perform worse than their non-reasoning counterparts on special rules—a pattern observed across all domains.

On normal rules, most models exceed 90% accuracy, indicating strong pattern recognition when the rule is surface-aligned or structurally obvious. In contrast, performance on special rules drops significantly. For example, in chess, non-reasoning models like GPT-4o and DeepSeek-V3 reach 55–65% on SR1, while their reasoning counterparts fall below 25%. Similar gaps appear in Texas Hold'em, Dice, and Blackjack (Fig. 2). Additional comparison results can be found in Appendix E.

These results show that reasoning models struggle more with exception-based or hidden rules. This suggests that their multi-step traces may not help—and can introduce incorrect assumptions or misleading intermediate steps. We examine this hypothesis in detail in RQ2 by analyzing the reasoning outputs directly.

> **RQ1: How well do LLMs perform on inductive reasoning tasks, and has this improved with recent models?**
> **Answer:** Non-reasoning LLMs consistently outperform reasoning-enabled models on inductive tasks with hidden rules, showing that recent reasoning strategies degrade performance on abstraction beyond surface patterns.

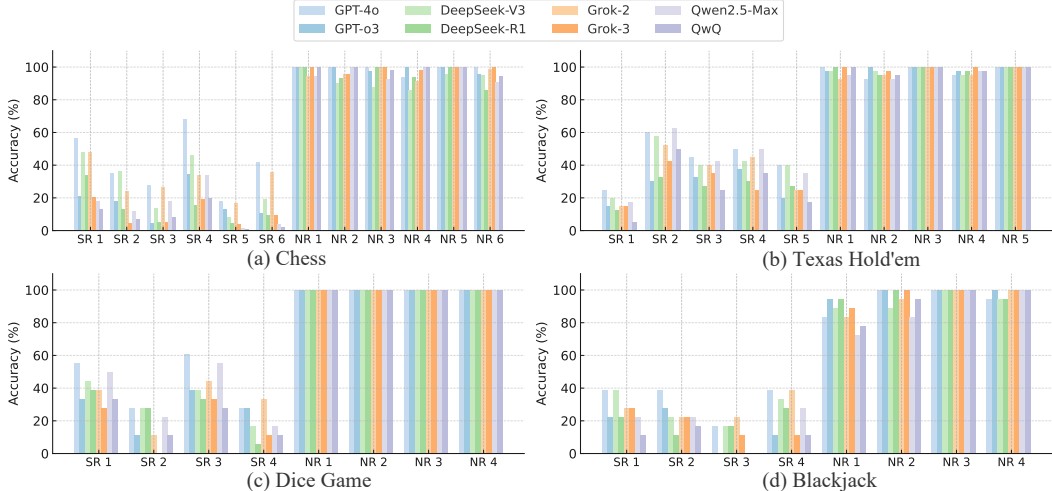

Figure 2: Inductive accuracy on normal rules (NRs) and special rules (SRs) across four games. Each bar shows rule-wise inductive performance for eight LLMs. While most models achieve high accuracy on NRs, reasoning models (lighter bars) consistently underperform non-reasoning models (darker bars) on SRs, indicating that current reasoning may hurt inductive abilities on hidden rules.

## 4 Root Causes of Reasoning Failures

To understand why reasoning sometimes fails to improve—and in some cases even harms—the inductive performance of LRMs (RQ2), we conduct both theoretical and empirical analyses to identify the root causes of these failures.

### 4.1 Theoretical Analysis of Reasoning Errors

To explain why reasoning can fail, we present a theoretical framework that models chain-of-thought reasoning as a sequence of discrete operations: *posing* a sub-task, *solving* it, and *summarizing* the final answer. Each step $k \in \mathbb{N}$ is associated with a reasoning state:

$$x_k = (m_k,\ s_k) \in \underbrace{\mathbb{R}^d}_{\text{belief state}} \times \underbrace{\{\text{NEEDQ}, \text{NEEDA}, \text{FINISH}\}}_{\text{reasoning mode}}, \tag{1}$$

where $m_k$ denotes the model's current belief about the correct answer $y^\star \in \mathbb{R}^d$, and $s_k$ tracks the stage of reasoning. Detailed mode semantics are provided in Appendix C.

**Evidence model.** The model does not observe the true target $y^\star$ directly. Instead, at each reasoning step $k$, it receives an indirect evidence vector $g_k$ based on its attempted sub-task resolution. We assume the evidence signal follows:

$$g_k = \alpha_k(y^\star - m_{k-1}) + \varepsilon_k, \qquad \varepsilon_k \sim \mathcal{N}(0, \sigma^2 I_d) \tag{2}$$

with the following interpretation:

- **Task alignment ($\alpha_k$).** The scalar $\alpha_k \in [-1, 1]$ represents how well the current sub-task focuses on the relevant latent structure. This captures an observed behavior in LLM reasoning: incorrect or ambiguous intermediate questions can derail downstream answers by misdirecting attention [17, 21]. Negative alignment ($\alpha_k < 0$) corresponds to misleading questions—such as proposing irrelevant concepts—which have been shown to increase failure rates in multi-hop reasoning [35, 36]. The scalar form abstracts this variable attention quality while keeping the model analytically tractable.
- **Answer noise ($\varepsilon_k$).** The residual $\varepsilon_k$ models stochastic variation in sub-task resolution, including token sampling noise, hallucinations, and unstable decoding paths. Prior work has empirically shown that even when CoT traces are well-posed, decoding introduces variability that degrades reliability [37, 38].

**Belief update and error propagation.** At each reasoning step, the model updates its internal belief by integrating a new evidence signal:

$$m_k = m_{k-1} + \gamma_k g_k, \tag{3}$$

where $m_k \in \mathbb{R}^d$ denotes the model's current belief about the target solution $y^\star$, and $\gamma_k \in (0, 1)$ is a step-size scalar that controls how much the new evidence shifts the belief. This form is justified by two observations.

- **Incremental refinement.** Empirical analyses show that LLMs often revise answers through local adjustments rather than full restatements [37]. The additive form captures this bounded update behavior, consistent with chain-of-thought reasoning where successive steps refine a hypothesis instead of discarding prior context.
- **Step-size heterogeneity.** The weight $\gamma_k$ reflects the model's implicit confidence, leading to varying step sizes [39, 40], as reasoning trajectories should not integrate all steps equally.

Subtracting $y^\star$ from both sides, we define the belief error $e_k := m_k - y^\star$ and obtain the recursion:

$$e_k = (1 - \gamma_k \alpha_k)e_{k-1} - \gamma_k \varepsilon_k, \tag{4}$$

**Emergence of reasoning errors.** Eq. (4) helps us to decompose the error into three error components:

**(1) Incorrect sub-task decomposition (Breakdown Error, $\alpha_k$).** Breakdown errors directly correspond to poor question alignment ($\alpha_k \approx 0$) or negative alignment ($\alpha_k < 0$). In such scenarios, the coefficient $(1 - \gamma_k \alpha_k)$ magnifies or maintains the previous error magnitude $\|e_{k-1}\|$, thus preventing error reduction or even causing divergence. (Detailed analysis can be found in Appendix F.)

**(2) Incorrect sub-task solving (Solving Error, $\varepsilon_k$).** Even under optimal question alignment ($\alpha_k \approx 1$), the inherent answer-generation noise $\varepsilon_k$ introduces stochastic deviations at each reasoning step. (Detailed analysis can be found in Appendix F.)

**(3) Incorrect final answer summarization (Summary Error).** The third class of reasoning error arises from deciding when to stop the reasoning process and commit to a final answer. This halting decision determines the total number of reasoning steps $N$, which directly affects the model's prediction $\hat{y}_N := m_N$. We analyze the resulting prediction error by computing the expected squared deviation from the true target $y^\star \in \mathbb{R}^d$. Let $e_N := m_N - y^\star$ denote the final error. Then the expected error is given by:

$$\mathcal{E}(N) = \mathbb{E}\|e_N\|^2 = b_0 \prod_{i=1}^{N}(1 - \gamma_i \bar{\alpha})^2 + \sigma^2 \sum_{i=1}^{N} \gamma_i^2 \prod_{j=i+1}^{N}(1 - \gamma_j \bar{\alpha})^2 + \Delta(N) \tag{5}$$

where $b_0 = \|m_0 - y^\star\|^2$ is the initial squared error, $\gamma_i \in (0, 1)$ is the integration weight at step $i$, and $\bar{\alpha} := \mathbb{E}[\alpha_k]$ is the expected question alignment. The product terms reflect accumulated error contraction across steps, and $\Delta(N) \geq 0$ accounts for additional variance due to misalignment variability. A full derivation of Eq. (5) is provided in Appendix G.

**Theorem 4.1** (Optimal Reasoning Length Exists). *Let the expected mean-squared error after $N$ Answer defined in Eq 5. Then:*

1. *$\mathcal{E}(N)$ is strictly decreasing for $N < N^\star$ and strictly increasing for $N > N^\star$, with*

$$N^\star := \min\{N \geq 0 : \mathcal{E}(N+1) > \mathcal{E}(N)\};$$

2. *the minimiser $N^\star$ is unique. Consequently, $\mathcal{E}(N)$ is U-shaped when plotted against reasoning depth $N$.*

Proofs can be found in Appendix G.4. The structure of Eq 5 can be intuitively interpreted to reveal a fundamental bias–variance trade-off. The first term shrinks as $N$ increases: repeated evidence integration reduces the bias. However, both the second term (arising from answer-generation noise) and the third term (from stochastic question quality) grow with $N$, reflecting variance accumulation. This competition creates a non-monotonic curve in $\mathcal{E}(N)$: at small depths, additional reasoning reduces error, but beyond a critical point, further reasoning begins to increase it. Reasoning that terminates at any depth $N \neq N^\star$ incurs additional error due to either under- or over-reasoning. We refer to this deviation as the *Summary Error*—a global error that arises not from a single misstep, but from a misjudged stopping point over an otherwise well-structured reasoning trajectory.

## 4.2 Theoretical Guidance for Intervention Design

In this section, we present additional theoretical results that clarify the structure of each component, which serves the foundation of mitigation methods that discussed in Section 5. Full derivations are provided in Appendix G.

**Controlling sub-task decomposition error ($\alpha_k$).** The alignment scalar $\alpha_k$ determines how well the posed sub-question targets the true residual error. Positive values ($\alpha_k > 0$) reduce bias; negative values ($\alpha_k < 0$) increase it exponentially with reasoning depth. Lemma G.7 establishes that any improvement in alignment ($\alpha_k \uparrow$) strictly reduces error, while misaligned reasoning ($\alpha_k < 0$) guarantees divergence. This motivates interventions that eliminate unconstrained question posing, that we will discuss in Section 5.

**Controlling sub-task solving error ($\varepsilon_k$).** The noise term $\varepsilon_k$ models stochastic variation in answer generation, which is scaled by the belief integration weight $\gamma_k$, making $\gamma_k$ a critical amplifier of solving error. Lemma G.6 formalizes this relationship: while alignment ($\alpha_k$) always improves performance, there exists an optimal $\gamma_k^\star$ that balances the benefit of information integration against the risk of over-amplifying noise. This interaction motivates our intervention at the solving phase. We will reduce effective variance $\sigma^2$ by anchoring the model's outputs to structurally valid reasoning traces in Section 5.

**Controlling summarization error (global stopping rule).** Reasoning must terminate at some depth $N$, at which point the model commits to its final prediction $\hat{y}_N$. Theorem 4.1 proves that expected squared error $\mathcal{E}(N)$ is U-shaped in $N$, with a unique optimum $N^\star$. In addition, Lemma G.5 provides a closed-form solution for $N^\star$ in the constant-$\gamma$ case, offering theoretical guidance on setting reasoning length. Accordingly, we constrain model outputs via a fixed token budget to discourage excessively long traces and nudges reasoning toward near-optimal depths in Section 5.

## 4.3 Empirical Analysis of Reasoning Errors

We empirically validate the theoretical taxonomy introduced in Section 4.1, which identifies three primary sources of reasoning failure: **(1) Incorrect Sub-task Decomposition (Breakdown Error)** arising from incorrect sub-task decomposition, **(2) Incorrect Sub-task Solving (Solving Error)** caused by noise in sub-task resolution, and **(3) Incorrect Final Answer Summarization (Summary Error)** resulting from premature or excessive reasoning steps. These failure modes are grounded in the dynamics of the belief update equation (4), where errors propagate via suboptimal alignment ($\alpha_k$), additive noise ($\varepsilon_k$), and misjudged stopping time $N$.

Among these, **Solving Error dominates across all models and tasks**, accounting for over 80% of failure cases. While theoretically modeled as additive noise, solving failures often exhibit structured patterns in practice. Based on prior analyses of LLM reasoning drift Based on consistent error patterns we observed in over 100 failed reasoning traces across multiple tasks and models, we classify these errors into three observable subtypes: (1) *Math Overuse*, where models inappropriately apply arithmetic operations to symbolic inputs (e.g., card suits or chess pieces); (2) *Overgeneralization*, where rules are inferred from few examples without proper validation; and (3) *Hallucinated Rules*, where fabricated constraints are introduced without support from input observations. Representative examples for each are provided in Appendix I.

Breakdown Errors are less frequent but still consequential, especially in structurally complex games like Texas Hold'em. These correspond to misaligned sub-task decomposition, where the model fixates on irrelevant features or ignores core inductive structure. Summary Errors are the least frequent and occur when models produce overly long or overly short reasoning chains, diverging from the optimal depth $N^\star$ identified in Theorem 4.1.

To ensure reliability, each failure trace was independently reviewed by two annotators using shared labeling criteria (Detailed in Appendix K). While some interpretation was involved—as is typical in reasoning error analysis—agreement was high, and disagreements were resolved collaboratively. Rather than impose rigid boundaries, we aim to capture recurring, interpretable failure patterns observed consistently across models and games. Table 1 summarizes the distribution, and Appendix H, I, and J provides representative examples for each error type.

Three key observations emerge: First, **Solving Errors dominate**, accounting for over 80% of failures in most settings. This aligns with our theoretical model, where errors introduced during sub-task

Table 1: Error rate analysis across different games and models

| Games | Models | Error Rate (count / total) | | | | |
|-------|--------|---------------------------|---|---|---|---|
| | | Breakdown | Solving | | | Summary |
| | | | Hallucinated Rule | Overgeneralization | Math Overuse | |
| Chess | DeepSeek-R1 | 5.8% (64/1109) | 17.2% (191/1109) | 22.3% (247/1109) | 47.4% (526/1109) | 7.3% (81/1109) |
| | QwQ | 4.3% (52/1217) | 16.5% (201/1217) | 26.1% (318/1217) | 52.1% (634/1217) | 1.0% (12/1217) |
| | Grok3 | 4.1% (55/1333) | 14.5% (193/1333) | 24.5% (327/1333) | 50.7% (676/1333) | 6.2% (82/1333) |
| Texas Hold'em | DeepSeek-R1 | 9.3% (14/151) | 15.9% (24/151) | 11.9% (18/151) | 58.9% (89/151) | 4.0% (6/151) |
| | QwQ | 14.0% (21/150) | 15.3% (23/150) | 22.0% (33/150) | 42.7% (64/150) | 6.0% (9/150) |
| | Grok3 | 13.2% (19/144) | 26.4% (38/144) | 12.5% (18/144) | 40.3% (58/144) | 7.6% (11/144) |
| Dice games | DeepSeek-R1 | 5.7% (3/53) | 11.3% (6/53) | 20.8% (11/53) | 60.4% (32/53) | 1.9% (1/53) |
| | QwQ | 10.3% (4/39) | 12.8% (5/39) | 23.1% (9/39) | 48.7% (19/39) | 5.1% (2/39) |
| | Grok3 | 3.4% (2/59) | 18.6% (11/59) | 10.2% (6/59) | 61.0% (36/59) | 6.8% (4/59) |
| Blackjack | DeepSeek-R1 | 6.7% (4/60) | 21.7% (13/60) | 13.3% (8/60) | 53.3% (32/60) | 5.0% (3/60) |
| | QwQ | 8.6% (6/70) | 20.0% (14/70) | 28.6% (20/70) | 40.0% (28/70) | 2.9% (2/70) |
| | Grok3 | 8.2% (5/61) | 18.0% (11/61) | 19.7% (12/61) | 47.5% (29/61) | 6.6% (4/61) |

resolution ($\varepsilon_k$) propagate through the reasoning chain. Second, within Solving Errors, **Math Overuse consistently ranks highest**. For example, it accounts for 60.4% of all failures in Dice Games (DeepSeek-R1) and 53.3% in Blackjack. This suggests a recurring bias where models incorrectly apply arithmetic operations to symbolic domains—e.g., treating card suits or piece positions as numeric inputs. Notably, this pattern holds across structurally diverse tasks and models, indicating a deeper inductive misalignment rather than task-specific noise. Third, **Breakdown Errors peak in Texas Hold'em** (up to 14.0%), reflecting its higher structural complexity. In contrast, **Summary Errors remain rare** across the board (<8%), implying that most failures arise during solving rather than early decomposition or final summarization. Together, these results indicate that reasoning fails not from lack of logical capacity, but because early inductive errors—especially Math Overuse—are preserved and propagated across multi-step reasoning chains.

---

**RQ2: Why does reasoning sometimes fail to improve inductive performance in large language models?**
**Answer.** Reasoning propagates error: when sub-task decomposition is misaligned, or solving introduces noise, each reasoning step compounds the mistake. Such failures dominate in practice, making deeper reasoning harmful unless each step is reliable.

---

## 5 Improving CoT Reasoning

### 5.1 Intervention Design

Building on our theoretical framework, we design interventions that directly target each of the three failure modes: incorrect sub-task decomposition, sub-task solving noise, and overextended reasoning. The Appendix L shows the detailed prompts for each intervention.

**Sub-task decomposition.** To address Breakdown Error associated with misaligned sub-task formulation ($\alpha_k \approx 0$), we replace free-form reasoning with structured decomposition templates. Each template explicitly separates the reasoning into three phases: (i) identifying relevant entities in the input (e.g., cards, pieces, or dice), (ii) inducing candidate rules based on observed patterns, and (iii) verifying whether new cases satisfy those rules. This design constrains model output to stay within task-relevant abstractions and improves alignment between sub-questions and the global task objective. Prior work has shown that semantically aligned decompositions improve reliability in multi-step reasoning [11].

**Sub-task solving.** Solving Error arises when models introduce stochastic variation ($\varepsilon_k$) even under good decomposition. The most frequent subtype is Math Overuse, where models impose numeric operations onto symbolic inputs. To mitigate this, we follow Kuo et al. [41] and guide solving with worked examples that avoid numeric extrapolation. These examples anchor model behavior to structurally relevant patterns, reducing variance and discouraging inappropriate generalization.

**Answer summarization.** Summary Error occurs when reasoning continues past the optimal step count $N^\star$, causing error accumulation from unnecessary updates. We adopt recent ideas from efficient reasoning [42, 43] and impose a strict token budget of 1000 tokens per instance. This constraint limits excessive generation and encourages early commitment to accurate conclusions.

**Combined intervention.** Each intervention targets a distinct error component, but failure modes often co-occur. We therefore evaluate the combined setting, which integrates decomposition templates, solving-phase examples, and summarization constraints to stabilize the entire reasoning traces.

## 5.2 Results and Analysis

We evaluate the improving of performance guided by our intervention strategies across all four games, measuring rule-level induction accuracy under each condition. The results can be found in Fig. 3.

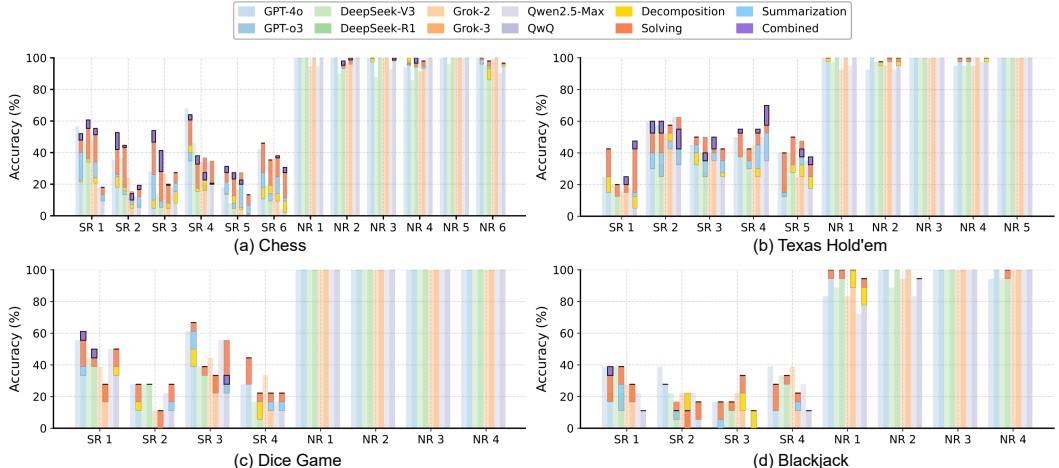

Figure 3: Inductive rule accuracy across different intervention strategies and models for each game domain. Each subfigure corresponds to one game; bars show average rule-wise accuracy under different reasoning-stage interventions. Across all domains, combined intervention (rightmost bars) achieves the highest performance, especially on special rules (SRs), indicating that structured decomposition, guided solving, and summarization control jointly enhance inductive abilities.

We observe consistent improvements under the combined intervention setting, especially on SRs, which involve higher inductive complexity. In most cases, combined intervention leads to clearly higher SR accuracy than any individual strategy. For example, in Chess and Dice, SR1–SR3 accuracy increases by 20–40% when guided chain-of-thought is applied, suggesting that multiple types of reasoning failures often co-occur and need to be addressed together. Compared to non-reasoning models, the combined intervention also achieves higher accuracy on both SRs and NRs across most games, indicating that improvements come from better reasoning structure rather than longer reasoning steps. Among individual strategies, sub-task decomposition is most helpful in structurally rich games like Chess and Blackjack, while solving-stage guidance contributes more in symbol-heavy domains such as Dice and Texas Hold'em. Summarization control provides moderate improvements across all tasks by reducing over-generation in the final answer stage. Performance on NR rules remains saturated under all settings, confirming that gains on SRs are not a result of trade-offs with simpler rules. However, we also observe that combined intervention does not always outperform the best individual strategy. In some cases, interactions between decomposition and solving stages may limit overall effectiveness, highlighting the difficulty of coordinating multiple reasoning processes. Detailed results for interventions are provided in Appendix M.1, and we further discuss how these interventions may generalize to more common reasoning problems in Appendix M.2.

> **RQ3: How can we improve the inductive performance of reasoning-enabled LLMs?**
> **Answer.** Inductive performance improves when reasoning is constrained. We achieve consistent gains by (i) enforcing structured decomposition, (ii) guiding solving with non-numeric examples, and (iii) limiting over-generation through token budgets. Combined, these interventions reduce error amplification across all reasoning phases.

# 6   Conclusion

In this paper, we investigate the inductive abilities of large language models through four designed games. Our analysis reveals that, contrary to expectations, reasoning—particularly in LRMs—does not always help; when poorly structured, it can even harm performance by introducing irrelevant or misleading chains of thought. We theoretically identify three core types of reasoning failure and empirically verify them through human evaluation of model-generated traces. To address these issues, we introduce targeted interventions at the decomposition, solving, and summarization stages. Experimental results show that each intervention independently improves inductive accuracy, confirming the importance of structured, well-controlled reasoning.

## Acknowledgement

This research was supported by the National AI Research Resource (NAIRR) Pilot NAIRR240283.

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

# A Details for Judging

## A.1 Examples of Judging

To illustrate the evaluation process, we provide examples of GPT-4o judging whether a model's induced rule aligns with the ground-truth. For each case, we show (1) the true rule, (2) the model-induced rule, and (3) GPT-4o's binary judgment.

Table 2: Examples of GPT-4o judging model-induced rules. Each row shows the true rule, the model's induced rule, and GPT-4o's binary alignment judgment with explanation.

| Game | Ground-Truth Rule | Model-Induced Rule | GPT-4o Judgment |
|---|---|---|---|
| Chess | SR6: Move in a straight line any number of squares, followed by a one-square diagonal shift. | The piece moves straight and then one square diagonally. | Yes, the hypothesis matches the ground-truth. The structure and intent are preserved. |
| Chess | SR4: Jump to the symmetric position across the nearest blocking piece. | The piece teleports to a random mirrored position. | No, the hypothesis does not match. The true rule requires a specific symmetry across blockers. |
| Texas Hold'em | SR3: Alternating odd–even values are treated as a mirror hand, beating any straight. | A hand with alternating red and black suits is promoted to beat any straight. | No, the hypothesis does not match. The ground-truth is based on numeric parity, not suit color. |
| Texas Hold'em | SR4: Five consecutive even numbers in the same suit count as a straight flush. | A hand with five even cards, regardless of suit, is treated as a flush. | No, the hypothesis does not match. It ignores the suit constraint and downgrades the hand. |
| Dice Game | SR1: If the sum is a prime number, the roll beats all hands including triples. | If the sum is prime, the roll wins. | Yes, the hypothesis matches the ground-truth. The paraphrasing is correct. |
| Dice Game | SR4: A pair plus a third die differing by exactly one (e.g., 4–4–5) beats regular pairs. | If a roll contains a pair and any third die, it beats all totals. | No, the hypothesis does not match. It omits the "difference of one" condition. |
| Blackjack | SR2: A three-card straight flush is treated as a blackjack. | If a hand has three consecutive cards in the same suit, it is equivalent to blackjack. | Yes, the hypothesis matches the ground-truth. The core condition is preserved. |
| Blackjack | SR3: A pair of different suits causes automatic loss. | If a hand contains a pair, it loses automatically. | No, the hypothesis does not match. It generalizes beyond the suit constraint. |

## A.2 Human Evaluation for Judging

To address potential circularity in using GPT-4o as a judge, we conducted a human validation study on 100 rule descriptions generated by GPT-o3 across all four games. Three independent human annotators, blind to both the model identity and GPT-4o's original decisions, assessed whether each generated description was semantically aligned with the ground-truth rule. We then compared the majority human judgment with GPT-4o's evaluations and computed Cohen's $\kappa$ to quantify agreement.

Table 3: Human validation of GPT-4o's judgments on rule alignment.

| Judgment Type | Human Majority | GPT-4o | Overall Accuracy | Cohen's $\kappa$ |
|---|---|---|---|---|
| Aligned Cases | 60 | 60 | 1.00 | 1.00 |
| Not Aligned Cases | 40 | 40 | 1.00 | 1.00 |
| **Overall** | **100** | **100** | **1.00** | **1.00** |

GPT-4o's assessments matched the human in all 100 cases (60 aligned and 40 not aligned), yielding an accuracy and $\kappa$ of 1.0. While this perfect agreement may partially reflect the limited sample size, it provides evidence that GPT-4o's evaluations are well-aligned with human judgment in this context.

## A.3 Prompts for Judging

To evaluate whether a model-inferred rule matches the ground-truth, we prompt GPT-4o with the following template. The input includes the game context, the correct rule, the model hypothesis, and a fixed binary instruction.

GPT-4o Judgment Prompt

You are given a reasoning task involving [GAME TYPE].
Below is the ground-truth rule and a model's hypothesis.
Decide whether the hypothesis matches the ground-truth rule in semantics.
Ground-truth rule: [INSERT TRUE RULE]
Model-induced rule: [INSERT MODEL RULE]
Answer with: "Yes, the induced rule matches the ground-truth." or "No, the induced rule does not match." Briefly explain your reasoning in one sentence.

# B  Full Rule Specifications for Each Game

## B.1  Chess

We define eight types of pieces. Each is assigned one rule from the following pools.

**Normal Rules (NRs):**

- NR1: Move one square in any direction.
- NR2: Move in an L-shaped pattern: two squares in one direction and one square perpendicular.
- NR3: Move any number of squares diagonally.
- NR4: Move exactly two squares forward (in the direction of increasing row).
- NR5: Move any number of squares straight (horizontally or vertically).
- NR6: Move exactly two squares diagonally.

**Special Rules (SRs):**

- SR1: Move in a straight line any number of squares, then shift vertically by exactly two squares.
- SR2: Move diagonally any number of squares, then two squares in a perpendicular diagonal direction.
- SR3: Move exactly three squares in one direction, then move one square downward.
- SR4: Jump to the symmetric position across the nearest blocking piece.
- SR5: Swap with a target piece on an occupied square within distance $\leq 3$.
- SR6: Move in a straight line any number of squares, followed by a one-square diagonal shift.

Each game selects four NRs and four SRs, randomly assigning one rule to each piece. Rule assignments are reshuffled every episode.

## B.2  Texas Hold'em

**Normal Rules (NRs):**

- NR1: A hand with one pair is treated as stronger than any high card.
- NR2: A hand with three of a kind is treated as stronger than two pairs.
- NR3: A straight (five cards in sequential rank, any suit) is treated as stronger than three of a kind.
- NR4: A flush (five cards of the same suit, not in sequence) is treated as stronger than any straight.
- NR5: Four of a kind (four cards of the same rank) is treated as stronger than any flush.

**Special Rules (SRs):**

- SR1: A hand containing five consecutive prime numbers (e.g., 2–3–5–7–J) is treated as stronger than any three-of-a-kind.
- SR2: A hand with alternating card colors (e.g., red–black–red–black–red) is treated as a straight regardless of numeric order.
- SR3: A hand with alternating odd and even values is treated as a "mirror hand" and beats any straight.
- SR4: A hand containing five consecutive even numbers in the same suit is treated as a straight flush.
- SR5: A hand with four cards of one parity (odd/even) and one of the opposite parity is treated as a "hybrid hand," ranking just below four of a kind.

Games sample two NRs and two SRs per episode, covering 100 combinations ($C_5^2 \times C_5^2$), each with twelve hands (four per rule).

## B.3 Dice Games

**Normal Rules (NRs):**

- NR1: A total sum between 4 and 10 (inclusive) is a "small total."
- NR2: A total sum between 11 and 17 (inclusive) is a "large total."
- NR3: A roll containing any pair is treated as stronger than small or large totals.
- NR4: A triple (three identical dice) is treated as stronger than any pair or total.

**Special Rules (SRs):**

- SR1: If the total sum is a prime number, the roll beats any hand including triples.
- SR2: If all three dice are prime numbers (2, 3, 5), the roll beats all hands except SR1.
- SR3: If the dice alternate in parity (odd–even–odd or even–odd–even), the roll beats all hands except SR1/SR2/triples.
- SR4: If the roll contains a pair and the third die differs from the pair by exactly one (e.g., 4–4–5), the roll beats any regular pair or total.

36 rule combinations ($C_4^2 \times C_4^2$), 12 observations per episode, 18 appearances per rule.

## B.4 Blackjack

**Normal Rules (NRs):**

- NR1: A hand totaling exactly 21 is a "blackjack" and wins.
- NR2: Any hand exceeding 21 is a bust. If both bust, the closer total to 21 wins.
- NR3: If neither busts nor hits 21, the hand with the higher total wins.
- NR4: An ace can be counted as either 1 or 11 to optimize the hand.

**Special Rules (SRs):**

- SR1: If the total sum is a prime number, the hand wins regardless of bust.
- SR2: A three-card straight flush is treated as a "blackjack" regardless of total.
- SR3: A hand with exactly one pair of different suits is a special loss.
- SR4: A hand with three non-consecutive values where the middle equals the average of the other two (e.g., 3–6–9) is an automatic win.

Each hand contains five cards. Rule combinations: $C_4^2 \times C_4^2 = 36$, 12 hands per episode, 18 per rule.

# C   Mode Semantics and Admissible Actions

At each step $k$, the model is in mode $s_k \in \{\text{NEEDQ}, \text{NEEDA}, \text{FINISH}\}$ with transitions:

$$
\begin{aligned}
s_k = \text{NEEDQ} &\Rightarrow a_k = \textbf{Ask}(q_k), \quad s_{k+1} = \text{NEEDA} \\
s_k = \text{NEEDA} &\Rightarrow a_k \in \{\textbf{Answer}, \textbf{Finish}\}, \quad s_{k+1} \in \{\text{NEEDQ}, \text{FINISH}\} \\
s_k = \text{FINISH} &\Rightarrow \text{halt; output } \hat{y} = m_k
\end{aligned}
$$

This structure enforces a cognitively meaningful control flow and localizes reasoning failure by step type.

# D   Prompts for LLMs to Induce Rules

## D.1   Prompts

The following prompt is used to instruct the model to induce latent rules from gameplay observations. Each instance provides a full episode of actions and outcomes. The model must identify the underlying rule that best explains the observed behavior.

**(Background Information)**
You are an agent tasked with discovering how a certain game works. The rules of the game are unknown to you in advance. However, you are given a full gameplay transcript from a single episode, showing the actions taken and whether they resulted in success or failure. The rule used in each episode is internally consistent, but may differ across episodes. Your goal is to infer this rule by observing what patterns emerge in the examples. **(Instruction)** Carefully analyze the episode shown below. Use only the observed behavior and outcomes to deduce the underlying rule. Be precise and concise. Your response should summarize the rule as clearly as possible, in natural language. Do not speculate beyond the given examples.
Do not list the examples again. Do not include uncertainty statements ("it might be...", "perhaps..."). Just state the rule that best explains the episode.
**(...insert game transcripts here...)**
**(Output Format)** Your response should follow this format:
Induced Rule: <your concise natural language description>

## D.2 Transcripts of Each Game

To support rule induction, each model is provided with a structured transcript capturing the observable behavior of a single game episode. These transcripts serve as the sole source of information available to the model—they do not include rule annotations or symbolic summaries. Instead, models must infer the governing logic by analyzing regularities in the recorded actions and outcomes.

Each game type has a customized transcript format reflecting its structural characteristics. For example, chess episodes log initial piece placements, move sequences, and capture events; Texas Hold'em transcripts include player hands, public cards, and final winners; dice games record raw rolls and outcome labels; blackjack tracks hands, point totals, and bust states.

Table 4 summarizes the key fields recorded for each domain, along with representative examples. These transcripts constitute the model's input during the rule induction task described in Section D.1.

Table 4: Gameplay transcript fields and structured examples for each game domain.

| Game | Recorded Fields per Episode | Example Entry |
|---|---|---|
| Chess | • Board size (e.g., 8×8, 15×15)
• Initial piece placement (e.g., Red King @ m14)
• Round-wise moves: source → target
• Events: captures, illegal moves
• Final outcome (optional) | • Board: 15×15
• Red King @ m14, Black Queen @ k2
• Round 1: Red: m14→o13; Black: k2→k0
• Red captures Black Bishop |
| Texas Hold'em | • Player hole cards (2 per player, with suits)
• Community cards (flop, turn, river)
• Winning player and hand rank
• Hand type comparison | • Player A: 2♠, 4♣
• Player B: 3♣, 3♠
• Board: 4♣, 5♥, 6♠, 9♠, Q♣
• Winner: Player B (Pair of Threes) |
| Dice Game | • Roll result: list of 3 dice
• Outcome: win/loss
• Optional: derived features (e.g., parity, sum, triple) | • Roll 1: [4, 4, 5] → Win
• Roll 2: [2, 6, 6] → Win
• Roll 3: [1, 3, 6] → Lose |
| Blackjack | • Player hand (5 cards)
• Total score after ace adjustment
• Whether bust occurred
• Outcome: win/loss/tie | • Player: 5♠, 3♣, A♣, 2♠, 9♥
• Total: 20 (A=11), No bust
• Result: Win vs Dealer (Bust) |

# E   Additional Inductive Accuracy

## E.1   Additional Inductive Accuracy across Model Versions and Sizes

To complement the main paper's results, we further investigate whether the performance gap between reasoning and non-reasoning models may be confounded by differences in model ver-

sions and sizes. To mitigate these factors, we include additional comparisons using the most recent publicly available reasoning model, `DeepSeek-R1-0528`, and its closest non-reasoning counterpart, `DeepSeek-V3-0325`. We also compare `GPT-o1` (`o1-2024-12-17`) against `GPT-4o` (`gpt-4o-2024-08-06`), as these represent adjacent releases with minimal temporal and architectural differences. For completeness, we also report the corresponding results from the main paper for fair comparison. Experiments are conducted on the **Chess** game, and the results are summarized in Table 5.

Table 5: Inductive accuracy on Normal Rules (NRs) and Special Rules (SRs) across different model versions and sizes on the Chess game.

| Models | SR1 | SR2 | SR3 | SR4 | SR5 | SR6 | NR1 | NR2 | NR3 | NR4 | NR5 | NR6 |
|---|---|---|---|---|---|---|---|---|---|---|---|---|
| DeepSeek-V3 (0325) | 48.00% | 48.00% | 14.00% | 46.00% | 8.00% | 19.33% | 100.00% | 90.00% | 88.00% | 86.00% | 96.00% | 95.33% |
| DeepSeek-R1 (0120) | 34.00% | 13.33% | 5.33% | 15.33% | 4.67% | 9.33% | 100.00% | 93.33% | 100.00% | 94.00% | 100.00% | 86.00% |
| DeepSeek-R1 (0528) | 38.00% | 17.33% | 8.00% | 18.00% | 8.00% | 11.33% | 100.00% | 100.00% | 96.00% | 95.33% | 100.00% | 91.33% |
| GPT-4o | 56.67% | 35.33% | 28.00% | 68.00% | 18.00% | 42.00% | 100.00% | 100.00% | 100.00% | 94.00% | 100.00% | 100.00% |
| GPT-o1 | 19.33% | 18.67% | 8.00% | 30.00% | 13.33% | 12.00% | 100.00% | 100.00% | 98.00% | 100.00% | 100.00% | 97.33% |
| GPT-o3 | 21.33% | 18.00% | 4.67% | 34.67% | 13.33% | 10.67% | 100.00% | 100.00% | 97.33% | 100.00% | 100.00% | 96.00% |

We notice that, when controlling for model family and release proximity, reasoning-augmented models continue to underperform their non-reasoning models on special rule (SR) generalization, while both perform similarly on normal rules (NR). This aligns with our core claim that reasoning can impair inductive generalization under hidden rule settings.

## E.2 Comparison Against Other Reasoning Methods

To further test inductive accuracy under different reasoning settings, we conducted additional experiments applying a Chain-of-Thought (CoT) style instruction to GPT-4o:

> *"Let's think step by step. First, extract the rule from the transcript. Then explain your reasoning."*

This instruction was applied to each rule inference task in the **Chess** game, and we compared model performance with and without CoT prompting; the results are in Table 6.

Table 6: Inductive accuracy of GPT-4o on Normal Rules (NRs) and Special Rules (SRs) with and without Chain-of-Thought (CoT) prompting on the Chess game.

| Methods | SR1 | SR2 | SR3 | SR4 | SR5 | SR6 | NR1 | NR2 | NR3 | NR4 | NR5 | NR6 |
|---|---|---|---|---|---|---|---|---|---|---|---|---|
| w/o CoT | 56.67% | 35.33% | 28.00% | 68.00% | 18.00% | 42.00% | 100.00% | 100.00% | 100.00% | 94.00% | 100.00% | 100.00% |
| w/ CoT | 58.00% | 34.00% | 27.33% | 68.00% | 20.00% | 42.00% | 100.00% | 100.00% | 100.00% | 97.33% | 100.00% | 100.00% |
| $\Delta$ (Change) | +1.33% | -1.33% | -0.67% | +0.00% | +2.00% | +0.00% | +0.00% | +0.00% | +0.00% | +3.33% | +0.00% | +0.00% |

Overall, CoT prompting did not lead to consistent improvements. On SR rules, performance was largely unchanged or slightly decreased, suggesting that added reasoning may not help—and can sometimes distract—from inferring hidden rules. On NR rules, performance was already saturated.

It is worth noting that CoT prompting here is fundamentally different from models like GPT-o3, which are explicitly trained or prompted to generate structured, multi-stage reasoning traces (e.g., decomposition $\rightarrow$ solving $\rightarrow$ summarization). Our findings indicate that simply injecting CoT into non-reasoning models does not reproduce the same failure patterns, nor does it resolve them.

## F Belief Update and Error Recursion

**Intuitive Goal.** Reasoning is modeled here as an *iterative information–integration process*: each **Ask** action decides *where to look next*, and each ensuing **Answer** returns *evidence* that nudges the current belief $m_{k-1}$ either closer to, or further from, the true solution $y^\star$. The agent does *not* observe $y^\star$ directly; instead it receives a vector signal that is *correlated* with the residual error $(y^\star - m_{k-1})$ but corrupted by two orthogonal noise sources:

1. **Question–alignment noise** ($\alpha_k$): cognitive or pragmatic imperfections in how the sub-question is framed. A well-posed question ($\alpha_k \approx 1$) elicits evidence that points

almost exactly along the residual error, whereas a misguided question ($\alpha_k \approx 0$) yields a useless tangent, and a catastrophically wrong question ($\alpha_k < 0$) returns evidence pointing the *opposite* way.

2. **Answer-generation noise** ($\varepsilon_k$): stochasticity in the LLM's sampling, retrieval errors, hallucinations, token truncation, or memory decay.

**Evidence Model.** Conditioned on the history up to step $k-1$, the **Answer** action produces the *evidence vector*

$$g_k \;=\; \underbrace{\alpha_k}_{\text{alignment}} \left(y^\star - m_{k-1}\right) \;+\; \underbrace{\varepsilon_k}_{\text{answer noise}} \;, \qquad \varepsilon_k \sim \mathcal{N}\!\left(0, \sigma^2 I_d\right), \tag{6}$$

where $\alpha_k \in [-1, 1]$ is an i.i.d. scalar with $\mathbb{E}[\alpha_k] = \bar{\alpha} \in (0, 1)$ and $\mathrm{Var}[\alpha_k] = \tau^2$. Eq. (6) embodies two critical properties:

1. *Directionality.* The deterministic component $\alpha_k(y^\star - m_{k-1})$ is collinear with the current error vector. Its amplitude encodes how well the sub-question aligns with the unknown residual structure.

2. *Zero-mean perturbations.* The random component $\varepsilon_k$ is isotropic and unbiased, reflecting that answer noise does *not* systematically drift the belief in any preferred direction.

**Belief–integration Weight** $\gamma_k$**.** Upon receiving $g_k$ the agent chooses how aggressively to incorporate it. The *step-integration weight* $\gamma_k \in (0, 1)$ captures resource–bounded cognitive fusion: large $\gamma_k$ trusts the new evidence, small $\gamma_k$ hedges against possible error. (learning rate of the belief state)

**Update Rule.** The new belief is therefore

$$m_k \;=\; m_{k-1} + \gamma_k \, g_k \;=\; m_{k-1} + \gamma_k \alpha_k \left(y^\star - m_{k-1}\right) + \gamma_k \varepsilon_k. \tag{7}$$

Subtracting $y^\star$ yields the *error recursion*

$$\begin{aligned} e_k &:= m_k - y^\star \\ &= \left(1 - \gamma_k \alpha_k\right) e_{k-1} \;-\; \gamma_k \, \varepsilon_k, \end{aligned} \tag{8}$$

which is linear, time-homogeneous, and driven by independent Gaussian shocks—exactly the form required for the closed-form analysis carried out in Appendix G.

**Relation to Classical Reasoning Notions.**

- The pair $(\alpha_k, \gamma_k)$ mirrors the cognitive psychology split between *question relevance* (task focus) and *processing depth* (how strongly new evidence is admitted into working memory).
- Eq. (7) can be interpreted as a *single–step Bayesian update* with a fixed-precision prior versus likelihood, or as a *stochastic approximation* à la Robbins–Monro.
- The recursion (8) is the reasoning-theoretic analogue of a *Kalman filter* on a static target with multiplicative observation noise.

# G  Closed-form Error Expression

## G.1  Unrolling the Recursion

Iterating the error recursion:

$$\begin{aligned} e_k &:= m_k - y^\star \\ &= \left(1 - \gamma_k \alpha_k\right) e_{k-1} \;-\; \gamma_k \, \varepsilon_k, \end{aligned} \tag{9}$$

for $N$ Answer steps yields

$$e_N \;=\; \left(\prod_{i=1}^{N}(1 - \gamma_i \alpha_i)\right) e_0 \;-\; \sum_{i=1}^{N}\left(\gamma_i \prod_{j=i+1}^{N}(1 - \gamma_j \alpha_j)\right)\varepsilon_i.$$

Denote $B_N := \prod_{i=1}^{N}(1 - \gamma_i \alpha_i)$, $b_0 := \|e_0\|^2$.

## G.2 Taking Expectations

Because the noises $\varepsilon_i$ are zero-mean, independent, and independent of $\{\alpha_i\}$, all cross terms vanish. Using $\mathbb{E}[\|\varepsilon_i\|^2] = \sigma^2$ we obtain

$$E(N) = B_N^2\, b_0 + \sigma^2 \sum_{i=1}^{N} \gamma_i^2 \Big( \prod_{j=i+1}^{N} (1 - \gamma_j \alpha_j) \Big)^2. \tag{10}$$

## G.3 Averaging over Question Alignment

Taking expectation w.r.t. $\alpha_i$ and writing $\bar{B}_N := \prod_{i=1}^{N}(1 - \gamma_i \bar{\alpha})$,

$$\mathbb{E}[E(N)] = \bar{B}_N^2\, b_0 + \sigma^2 \sum_{i=1}^{N} \gamma_i^2 \bar{B}_{i+1,N}^2 + \underbrace{\Delta(N)}_{= \ \tau^2\text{-order positive term}}, \tag{11}$$

where $\bar{B}_{i+1,N} := \prod_{j=i+1}^{N} (1 - \gamma_j \bar{\alpha})$ and $\Delta(N) \geq 0$ collects all contributions containing $\tau^2$.

**Interpretation.**

- *Bias contributor:* $\bar{B}_N^2\, b_0$ decreases monotonically with $N$.
- *Variance contributor:* the summation term grows strictly with $N$.
- *Mis-question variance $\Delta(N)$:* additional variance induced by imperfect question alignment, also monotonically increasing in $N$.

## G.4 Theorem: Inevitable U-shaped Error Curve

**Theorem G.1** (U-shape of expected error)**.** *Let the expected mean-squared error after $N$ **Answer** moves be*

$$\mathcal{E}(N) := b_0 \Big( \prod_{i=1}^{N}(1 - \gamma_i \bar{\alpha}) \Big)^2 + \sigma^2 \sum_{i=1}^{N} \gamma_i^2 \Big( \prod_{j=i+1}^{N} (1 - \gamma_j \bar{\alpha}) \Big)^2 + \Delta(N), \tag{12}$$

*where*

$$b_0 > 0, \quad 0 < \bar{\alpha} < 1, \quad 0 < \gamma_i \leq \bar{\gamma} < 1, \quad \sum_{i=1}^{\infty} \gamma_i = \infty,$$

*and where $\Delta(N) \geq 0$ is a non-decreasing sequence (the extra variance induced by question-alignment randomness $\tau^2$). Then:*

1. *$\mathcal{E}(N)$ is strictly decreasing for $N < N^\star$ and strictly increasing for $N > N^\star$, with*
$$N^\star := \min\{N \geq 0 : \ \mathcal{E}(N+1) > \mathcal{E}(N)\};$$

2. *the minimiser $N^\star$ is unique. Consequently, $\mathcal{E}(N)$ is U-shaped when plotted against reasoning depth $N$.*

*Proof.* Write (12) as

$$\mathcal{E}(N) = A(N) + V(N) + \Delta(N) \quad \text{with} \quad \begin{cases} A(N) := b_0\, P(N)^2, \\ V(N) := \sigma^2 \sum_{i=1}^{N} \gamma_i^2 P(i+1, N)^2, \end{cases}$$

where $P(a, b) := \prod_{j=a}^{b}(1 - \gamma_j \bar{\alpha})$ and we abbreviate $P(1, N)$ by $P(N)$.

**Step 1 (monotonicity of $A$ and $V$).** Because $0 < 1 - \gamma_i \bar{\alpha} < 1$, the product $P(N)$ decreases strictly as $N$ increases, hence $A(N)$ is strictly decreasing. For $V(N)$ observe

$$V(N+1) - V(N) = \sigma^2 \Big[ \gamma_{N+1}^2 + 2\sum_{i=1}^{N} \gamma_i^2 P(i+1, N)^2 \big( P(i+1, N+1)^2 - P(i+1, N)^2 \big) \Big] > 0,$$

since every summand is positive. Hence $V(N)$ is strictly increasing, and $\Delta(N)$ is non-decreasing by assumption.

**Step 2 (existence of a finite minimiser).** Because $\sum_i \gamma_i = \infty$, $P(N) \to 0$ and thus $A(N) \to 0$. Meanwhile $V(N)$ diverges to $+\infty$ and so does $\mathcal{E}(N)$. Therefore $\mathcal{E}(N)$ attains its minimum at some finite $N^\star$.

**Step 3 (uniqueness).** Define the forward difference $\Delta\mathcal{E}(N) := \mathcal{E}(N+1) - \mathcal{E}(N) = \big[A(N+1) - A(N)\big] + \big[V(N+1) - V(N)\big] + \big[\Delta(N+1) - \Delta(N)\big]$. The first bracket is negative, the latter two are positive. Because $A(N)$ decays geometrically while $V(N)$ grows unboundedly, there is exactly one index where the sign of $\Delta\mathcal{E}(N)$ flips from negative to positive; call it $N^\star$. Hence 1 and 2 follow. $\square$

**Remark G.2** (Interpretation of the U-shape)**.** *The term $A(N)$ captures residual bias: with every additional Answer step the multiplicative factor $(1 - \gamma_i\bar{\alpha})$ shrinks, so $A(N)$ falls exponentially. Conversely $V(N) + \Delta(N)$ aggregates variance: each step contributes fresh noise and never diminishes previous noise, so that component rises monotonically. Their competition enforces an optimum where further reasoning flips from net helpful to net harmful—producing the observed inverted-U curve for accuracy as depth increases.*

### G.5  Lemma: Closed-form Optimum When $\gamma_i \equiv \gamma$

**Lemma G.3** (Explicit $N^\star$ for constant step weight)**.** *Assume the integration weights are constant, $\gamma_i \equiv \gamma \in (0,1)$, and let $\rho := 1 - \gamma\bar{\alpha} \in (0,1)$. Further set $\tau^2 = 0$ so that $\Delta(N) = 0$. Then the expected error* (12) *reduces to*

$$\mathcal{E}_{geom}(N) \;=\; b_0\,\rho^{2N} \;+\; \sigma^2\gamma^2\,\frac{1 - \rho^{2N}}{1 - \rho^2}, \qquad N \in \mathbb{N}, \tag{13}$$

*and the unique minimiser $N^\star_{geom}$ is the smallest non-negative integer satisfying*

$$N \;\geq\; \frac{1}{2\,|\ln\rho|}\,\ln\!\Big(\frac{b_0\,(1 - \rho^2)}{\sigma^2\gamma^2}\Big). \tag{14}$$

*Consequently $\mathcal{E}_{geom}(N)$ is strictly decreasing for $N < N^\star_{geom}$ and strictly increasing for $N > N^\star_{geom}$.*

*Proof.* **Step 1 (geometric closed form).** With $\gamma_i \equiv \gamma$, $P(N) = \rho^N$. The variance series in (12) becomes a finite geometric sum:

$$V(N) = \sigma^2\gamma^2 \sum_{k=0}^{N-1} \rho^{2k} = \sigma^2\gamma^2\,\frac{1 - \rho^{2N}}{1 - \rho^2},$$

yielding (13).

**Step 2 (continuous relaxation).** Treat $N$ as a real variable and define $f(t) := b_0\rho^{2t} + \sigma^2\gamma^2(1 - \rho^{2t})/(1 - \rho^2)$ for $t \geq 0$. Differentiating,

$$f'(t) = 2(\ln\rho)\,\rho^{2t}\Big[-b_0 + \frac{\sigma^2\gamma^2}{1 - \rho^2}\Big].$$

Since $\ln\rho < 0$, the sign of $f'(t)$ flips exactly where the bracket vanishes, i.e. at $t^\star = \frac{1}{2|\ln\rho|}\ln\!\big(b_0(1 - \rho^2)/(\sigma^2\gamma^2)\big)$.

**Step 3 (integer minimiser).** Because $f(t)$ is strictly convex in $t$ (second derivative positive), its minimum over $\mathbb{N}$ is attained at the smallest integer not less than $t^\star$, which gives (14). The claimed monotonicity on either side of $N^\star_{geom}$ follows from the sign of $f'(t)$. $\square$

**Remark G.4** (Deterministic bias–variance balance)**.** *Equation* (13) *isolates the bias term $b_0\rho^{2N}$, decaying exponentially with depth, and the variance term $\sigma^2\gamma^2(1 - \rho^{2N})/(1 - \rho^2)$, growing monotonically from 0 to the asymptote $\sigma^2\gamma^2/(1 - \rho^2)$. The closed-form optimum* (14) *makes the bias–variance trade-off explicit: deeper reasoning is preferred when the initial misconception $b_0$ is large or the per-step noise $\sigma^2\gamma^2$ is small, and vice-versa. In all cases the curve remains U-shaped because the two components have opposite monotonicity with respect to $N$.*

**Lemma: Error behavior When $\alpha_i \equiv \alpha$**

**Lemma G.5** (U-shape with deterministic question alignment). *Fix a constant alignment factor $\alpha \in (0,1)$ and set $\tau^2 = 0$ so that $\Delta(N) = 0$ in (12). Let*

$$\mathcal{E}_\alpha(N) \;=\; b_0 \prod_{i=1}^{N}(1 - \alpha\gamma_i)^2 \;+\; \sigma^2 \sum_{i=1}^{N} \gamma_i^2 \prod_{j=i+1}^{N}(1 - \alpha\gamma_j)^2, \qquad N \in \mathbb{N},$$

*where the integration weights satisfy $0 < \gamma_i \le \bar\gamma < 1$ and $\sum_{i=1}^{\infty} \gamma_i = \infty$. Then:*

1. *There exists a finite $N_\alpha^\star$ such that $\mathcal{E}_\alpha(N+1) < \mathcal{E}_\alpha(N)$ for $N < N_\alpha^\star$ and $\mathcal{E}_\alpha(N+1) > \mathcal{E}_\alpha(N)$ for $N \ge N_\alpha^\star$.*

2. *The minimiser $N_\alpha^\star$ is unique, hence $\mathcal{E}_\alpha(N)$ is U-shaped in $N$.*

*Proof.* Define $P_\alpha(a, b) := \prod_{j=a}^{b}(1 - \alpha\gamma_j)$ and abbreviate $P_\alpha(1, N)$ by $P_\alpha(N)$. Decompose $\mathcal{E}_\alpha(N) = A_\alpha(N) + V_\alpha(N)$ with

$$A_\alpha(N) = b_0\, P_\alpha(N)^2, \quad V_\alpha(N) = \sigma^2 \sum_{i=1}^{N} \gamma_i^2\, P_\alpha(i+1, N)^2.$$

**(i) Monotonicity of $A_\alpha$.** Because $0 < 1 - \alpha\gamma_i < 1$, the product $P_\alpha(N)$ decreases strictly as $N$ grows; hence $A_\alpha(N)$ is strictly decreasing.

**(ii) Monotonicity of $V_\alpha$.** For any $N \ge 0$,

$$V_\alpha(N+1) - V_\alpha(N) = \sigma^2 \gamma_{N+1}^2 + \sigma^2 \sum_{i=1}^{N} \gamma_i^2 \big[P_\alpha(i+1, N+1)^2 - P_\alpha(i+1, N)^2\big] > 0,$$

because each bracket is positive. Thus $V_\alpha(N)$ is strictly increasing.

**(iii) Existence and uniqueness of optimum.** Since $\sum_i \gamma_i = \infty$, we have $P_\alpha(N) \to 0$, so $A_\alpha(N) \to 0$; but $V_\alpha(N) \to \infty$ because each term is non-negative and at least one term grows unbounded. Therefore the difference sequence $D(N) := \mathcal{E}_\alpha(N+1) - \mathcal{E}_\alpha(N)$ starts negative (bias dominates) and eventually becomes positive (variance dominates). Strict monotonicity of $A_\alpha$ and $V_\alpha$ implies $D(N)$ changes sign exactly once, establishing 1 and 2. □

**Remark G.6** (Bias–variance interpretation with fixed $\alpha$). *With deterministic question quality $\alpha$, every Answer move multiplies the residual bias by $1 - \alpha\gamma_i$, shrinking it monotonically. Variance still accumulates additively via the $\gamma_i^2$ term. Their opposing monotonicities enforce a single optimal reasoning depth $N_\alpha^\star$, beyond which additional steps corrupt more than they correct, regenerating the inverted-U pattern.*

## G.6 Lemma: Comparative Influence of Question Alignment ($\alpha$) versus Integration Weight ($\gamma$)

**Lemma G.7** (Marginal sensitivities for constant $\alpha$ and $\gamma$). *Assume the per-step parameters are deterministic and time-invariant,*

$$\gamma_i \equiv \gamma \in (0,1), \qquad \alpha_i \equiv \alpha \in (0,1), \qquad \tau^2 = 0,$$

*and define $\rho := 1 - \alpha\gamma \in (0,1)$. For a fixed reasoning depth $N \ge 1$ the expected error is (cf. Lemma G.5)*

$$\mathcal{E}_{\alpha,\gamma}(N) \;=\; b_0\, \rho^{2N} \;+\; \sigma^2 \gamma^2 \frac{1 - \rho^{2N}}{1 - \rho^2}.$$

*Then:*

1. ***Alignment is always beneficial.*** *The partial derivative of $\mathcal{E}_{\alpha,\gamma}(N)$ with respect to $\alpha$ is strictly negative:*

$$\frac{\partial}{\partial\alpha}\, \mathcal{E}_{\alpha,\gamma}(N) \;<\; 0, \qquad \forall\, (\alpha, \gamma) \in (0,1)^2,\ N \ge 1.$$

2. ***Integration weight exhibits a trade-off.*** *For fixed $\alpha$ the map $\gamma \mapsto \mathcal{E}_{\alpha,\gamma}(N)$ is U-shaped: there exists a unique $\gamma_N^\star \in (0,1)$ solving*

$$\frac{\partial}{\partial \gamma} \mathcal{E}_{\alpha,\gamma}(N) = 0, \tag{15}$$

*such that $\mathcal{E}_{\alpha,\gamma}(N)$ is strictly decreasing on $(0, \gamma_N^\star)$ and strictly increasing on $(\gamma_N^\star, 1)$.*

*Proof.* **(a) Monotonicity in $\alpha$.** Differentiate under the definition $\rho = 1 - \alpha\gamma$:

$$\frac{\partial \mathcal{E}}{\partial \alpha} = \frac{\partial \mathcal{E}}{\partial \rho}\frac{\partial \rho}{\partial \alpha} = (-\gamma)\frac{\partial \mathcal{E}}{\partial \rho}.$$

Because $\gamma > 0$, we only need $\partial\mathcal{E}/\partial\rho > 0$. Direct calculation gives

$$\frac{\partial \mathcal{E}}{\partial \rho} = 2N\,b_0\,\rho^{2N-1} + 2N\sigma^2\gamma^2\frac{\rho^{2N-1}}{1-\rho^2} > 0,$$

so the overall derivative is strictly negative, proving 1.

**(b) U-shape in $\gamma$.** Fix $\alpha$ and again write $\rho = 1 - \alpha\gamma$. Differentiate with respect to $\gamma$:

$$\frac{\partial \mathcal{E}}{\partial \gamma} = -2N\alpha b_0\,\rho^{2N-1} + 2\sigma^2\gamma\Big(\frac{1-\rho^{2N}}{1-\rho^2} - \frac{N\alpha\gamma\rho^{2N-1}}{1-\rho^2}\Big).$$

As $\gamma \downarrow 0$ the first (negative) term dominates, so $\partial\mathcal{E}/\partial\gamma < 0$; as $\gamma \uparrow 1$ the positive term proportional to $\sigma^2\gamma$ dominates, so the derivative is positive. Because the derivative is continuous on $(0,1)$ it crosses zero exactly once, defining the unique root $\gamma_N^\star$ that satisfies (15) and confers the claimed monotonicity.

Uniqueness of $\gamma_N^\star$ follows from strict convexity: $\partial^2\mathcal{E}/\partial\gamma^2 > 0$ on $(0,1)$ (direct but tedious algebra), ensuring the derivative changes sign only once. $\square$

**Remark G.8** (Practical reading). *Result 1 says that* better-focused questions *(higher $\alpha$) always* help, *regardless of noise or depth. Result 2 shows that* how aggressively one updates *($\gamma$) must be tuned: too timid ($\gamma \ll \gamma_N^\star$) leaves large residual bias, too bold ($\gamma \gg \gamma_N^\star$) inflates variance. Crucially, $\alpha$ influences only the* location *of the optimum via $\rho$, whereas $\gamma$ controls both* how fast bias falls *and* how steeply variance grows. *Thus, when per-step noise $\sigma^2$ is high, improving question alignment is the more reliable path to lower error; when noise is low, fine-tuning $\gamma$ around $\gamma_N^\star$ yields further gains.*

### G.7 Lemma: Qualitative Regimes of Question Alignment $\alpha$

**Lemma G.9** (Effect of $\alpha$ when $\gamma_i \equiv \gamma$). *Let the integration weight be a fixed $\gamma \in (0,1)$ and assume $\tau^2 = 0$. Define $\rho(\alpha) := 1 - \alpha\gamma$ and consider the expected error after $N \geq 1$* ***Answer*** *moves*

$$\mathcal{E}_\alpha(N) = b_0\,\rho(\alpha)^{2N} + \sigma^2\gamma^2\frac{1-\rho(\alpha)^{2N}}{1-\rho(\alpha)^2}. \tag{16}$$

*Then the qualitative behaviour splits into three disjoint regimes:*

1. ***Positive alignment*** $\left(0 < \alpha < \gamma^{-1}\right)$.
   *Here $0 < \rho(\alpha) < 1$ and*
   $$\frac{\partial}{\partial\alpha}\mathcal{E}_\alpha(N) < 0, \qquad \forall N \geq 1.$$
   *Increasing $\alpha$ always lowers* the error.

2. ***Zero alignment*** $(\alpha = 0)$.
   *Then $\rho = 1$ and $\mathcal{E}_0(N) = b_0 + N\sigma^2\gamma^2$. Bias never decays; variance grows linearly with depth.*

3. ***Negative alignment*** $\left(-\gamma^{-1} < \alpha < 0\right)$.
   *Now $\rho(\alpha) > 1$ and $\mathcal{E}_\alpha(N) = b_0\rho^{2N} + \sigma^2\gamma^2\frac{\rho^{2N}-1}{\rho^2-1}$, which increases strictly with $N$. Moreover $\frac{\partial}{\partial\alpha}\mathcal{E}_\alpha(N) > 0$; making $\alpha$ less negative* reduces error.

*Proof.* Differentiate (16) wrt. $\alpha$ via $\rho(\alpha)$:

$$\frac{\partial \mathcal{E}_\alpha}{\partial \alpha} = -\gamma \frac{\partial \mathcal{E}_\alpha}{\partial \rho}.$$

A direct calculation gives $\frac{\partial \mathcal{E}_\alpha}{\partial \rho} = 2N\rho^{2N-1}\left[b_0 - \frac{\sigma^2\gamma^2}{1-\rho^2}\right]$.

**Regime 1:** $0 < \rho < 1$. Because $1 - \rho^2 > 0$ we have $b_0 - \sigma^2\gamma^2/(1-\rho^2) < 0$, hence $\partial\mathcal{E}_\alpha/\partial\rho < 0$ and therefore $\partial\mathcal{E}_\alpha/\partial\alpha < 0$.

**Regime 2:** $\rho = 1$. Take the limit $\rho \uparrow 1$ in (16) to obtain the linear form stated.

**Regime 3:** $\rho > 1$. Now $1 - \rho^2 < 0$ so $b_0 - \sigma^2\gamma^2/(1-\rho^2) > 0$, making $\partial\mathcal{E}_\alpha/\partial\rho > 0$ and hence $\partial\mathcal{E}_\alpha/\partial\alpha > 0$. Moreover $\mathcal{E}_\alpha(N)$ increases with $N$ because both bias and variance terms grow when $\rho > 1$. $\square$

**Remark G.10** (Which lever matters more?). *An comparative discussion on alignment and integration weight.*

- ***Alignment*** *($\alpha$) is* unconditionally helpful *whenever it is positive—each incremental improvement shrinks error regardless of depth, noise, or weight. When alignment turns negative, performance deteriorates catastrophically with depth.*
- ***Integration weight*** *($\gamma$), by contrast, exhibits the bias–variance U-shape of Lemma G.6. It must be tuned around its optimal value $\gamma_N^\star$; deviations on either side hurt.*

*Therefore, ensuring* non-negative alignment *is the first priority—no choice of $\gamma$ can compensate for $\alpha < 0$—while fine-tuning $\gamma$ yields second-order gains once positive alignment has been secured.*

# H  Examples of Breakdown Error

Breakdown Errors occur when the model fails to properly decompose the task, leading to misaligned or irrelevant sub-questions.

---

**Breakdown Error Example**

**Game:** Chess
**Ground Truth:** Move in a straight line any number of squares, then shift vertically by exactly two squares.
**Model Reasoning:**
Let me break this down. First, I'll look at whether the piece lands on a square with a prime-numbered file or rank. That seems relevant, since many special rules involve numerical properties.
Next, I'll check if the move creates a diagonal alignment across pieces from both teams. If so, that might indicate a coordination-based rule.
The movement goes from b2 to e4. That's (+3 columns, +2 rows), which doesn't match a traditional L-shape or knight move, but perhaps it's a coded prime-sum rule?
The file total is b (2) + e (5) = 7, and 2 + 4 = 6. Hmm—maybe only prime-ranked destinations are allowed?
**Model Rule Answer:** The piece can move to squares where the file and rank add up to a prime number.
**Error:** The model's decomposition introduces irrelevant sub-questions (prime sums, team diagonals) that do not relate to the actual two-phase movement structure of SR1. Although the reasoning is internally coherent, it is grounded in a misaligned task interpretation. This constitutes a Breakdown Error: the model fails to recognize the structural form of the rule and instead explores symbolic numerical patterns unrelated to the task.

---

# I  Examples of Solving Error Subtypes

We provide representative reasoning traces for the three empirical subtypes of Solving Errors identified in our study. These traces illustrate how errors emerge during the solving stage, even when the

decomposition is valid. Each example shows the model's step-by-step reasoning and highlights where inductive drift or inappropriate generalization occurs.

## I.1 Hallucinated Rule

This subtype of Solving Error occurs when the model fabricates a rule that was never presented, demonstrated, or implied in the input data. Unlike overgeneralization—which extrapolates too broadly from limited evidence—hallucinated rules are entirely invented. These errors often emerge when the model tries to impose structure where none exists, or fills uncertainty with overconfident but unsupported patterns.

---

**Hallucinated Rule Example**

**Game:** Chess
**Ground Truth:** Move in a straight line any number of squares, followed by a one-square diagonal shift.
**Model Reasoning:**
(...earlier reasoning omitted...)
I'm thinking through how the piece might combine horizontal and vertical moves, akin to Brooklyn distance, to cover large distances effectively in a single turn.
It's interesting to think if the piece might follow an angular path, allowing it to shift vertically, horizontally, or diagonally in a single turn, adding a dynamic edge to the game.
(...continued reasoning...)
**Model Rule Answer:** The piece can move in a flexible angular trajectory that combines horizontal, vertical, and diagonal steps, as long as the total displacement approximates a straight path.
**Error:** The model hallucinates a continuous-space movement rule based on blended directional offsets. This rule does not exist in any defined NR or SR, and contradicts the stepwise constraint in SR6. The model invents a pseudo-geometry (e.g., "angular displacement") to rationalize incidental move patterns, reflecting a hallucinated inductive structure that misinterprets discrete board logic as continuous spatial flow.

---

## I.2 Overgeneralization

This subtype of Solving Error occurs when the model prematurely commits to a rule based on a small number of supporting examples. Rather than verifying consistency across diverse cases, it overextends an observed pattern to structurally dissimilar inputs. In our settings, overgeneralization often arises when models collapse multi-step actions into simplified summaries, or infer category-wide behavior from isolated instances. These errors reflect a tendency to form rules that are overly broad, failing to capture the precise structure or constraints intended by the task.

---

**Overgeneralization Example**

**Game:** Chess
**Ground Truth:** Move exactly three squares in one direction, then move one square downward.
**Model Reasoning:**
(...earlier reasoning omitted...)
Looking at this move from g4 to f7, the total shift is (-1 column, +3 rows). That's a net movement of three squares diagonally.
It looks similar to previous valid moves that landed three or four squares away from the starting point. So I assume this pattern allows 3-step diagonal moves.
Alternatively, maybe it's just a "distance-3" rule, where the end location must be offset by a vector of total length 3 or 4, in any direction. That would explain why g4 → f7 is allowed.
I'll go with that—this seems to fit with the movement pattern I've seen.
(...continued reasoning...)
**Model Rule Answer:** A piece can move 3 or 4 squares in any direction, including diagonals.
**Error:** The model collapses a composite 3+1 directional movement into a single displacement vector, overgeneralizing to a rule based on total distance. It ignores the two-phase structure and the downward constraint, falsely classifying structurally invalid moves as rule-consistent.

---

### I.3 Math Overuse

This error occurs when the model incorrectly applies arithmetic or numerical reasoning to tasks that are fundamentally symbolic, structural, or categorical in nature.

---

**Math Overuse Example**

**Game:** Chess
**Ground Truth:** Move in a straight line any number of squares, then shift vertically by exactly two squares.
**Model Reasoning:**
(...earlier reasoning omitted...)
OK, let me see. The piece's moves are quite varied, with differences like (2,8) and (2,11). This suggests they must stay within a maximum distance, maybe determined by the sum or difference of steps.
I'm analyzing the Euclidean distance for a piece's movement from b1 to i3, with column and row differences, leading to a distance of approximately 7.28. This method allows for accurate computation.
(...continued reasoning...)
**Model Rule Answer:** The piece can move to any square within a Euclidean distance of approximately 8 from its starting position.
**Error:** The model misinterprets the piece's movement by applying Euclidean distance as a decision criterion, which is not relevant to the rule-based nature of chess. This reflects a Math Overuse error: numerical metrics are substituted for structured reasoning over discrete directions and board geometry.

---

## J   Examples of Summary Error

Summary Errors occur when the model's reasoning process is internally consistent and correctly processes the input, yet the final rule it outputs is incorrect. Unlike Solving or Breakdown Errors, the failure here does not lie in intermediate logic or sub-task handling, but rather in the concluding inference step. These errors often manifest as misstatements, mislabelings, or minor structural inaccuracies in the final rule summary—despite evidence that the correct pattern was already identified.

---

**Summary Error Example**

**Game:** Chess
**Ground Truth:** The piece follows SR6: "Move in a straight line any number of squares, followed by a one-square diagonal shift."
**Model Reasoning:**
**Thinking 5m12s**
(...earlier reasoning omitted...)
Step 1: The move from c3 to f4 involves a three-square horizontal move (c → f), followed by a one-square diagonal shift upward and right (from row 3 to 4 and f to g).
Step 2: This clearly breaks down into: → A straight-line segment: c3 → f3 (horizontal, 3 squares), → Followed by: f3 → g4 (one-square diagonal shift). The diagonal is exactly one square and continues smoothly from the initial segment.
Step 3: The structure matches previous valid cases. The piece first makes a linear move, then pivots diagonally once. There are no blockers. The transition is clean.
(...reasoning ends...)
**Model Rule Answer:** The piece moves in a straight line followed by a vertical shift of one square.
**Error:** The model's reasoning accurately captures the two-stage movement required by SR6, identifying both the straight-line component and the one-square diagonal shift. However, it misstates the final rule, replacing "diagonal shift" with "vertical shift" in its summary. This is a Summary Error: the model reasons correctly but fails in final rule articulation.

---

# K  Classification of Reasoning Failures

To assess the reliability of our manual classification of reasoning failure types, we conducted an inter-annotator agreement study involving two independent annotators. Each annotator labeled 100 sampled outputs from reasoning models before any discussion or resolution of disagreements.

Table 7: Inter-annotator agreement between two annotators across three reasoning error types.

| Failure Mode | Agreement Count | Total Cases | Agreement Rate | Cohen's $\kappa$ |
|---|---|---|---|---|
| Decomposition Error | 36 | 40 | 0.900 | 0.84 |
| Solving Error | 38 | 42 | 0.905 | 0.87 |
| Summarization Error | 16 | 18 | 0.889 | 0.83 |
| **Overall** | **90** | **100** | **0.900** | **0.86** |

We computed Cohen's $\kappa$ between the two annotators based on their independent annotations prior to reconciliation. Agreement statistics for each error type are summarized in Table 7. The results indicate consistently high inter-annotator agreement across all categories, with overall $\kappa = 0.86$, suggesting strong reliability in the identification and classification of reasoning failure modes.

# L  Intervention Prompt Design

We provide full prompt templates used for each intervention strategy described in Section 5.1. These prompts were designed to isolate or jointly address the three major reasoning failure modes.

## L.1  Sub-task Decomposition Constraints

To mitigate Breakdown Errors caused by poorly aligned sub-question formulation, we provide structured prompts that explicitly constrain how models decompose the task. The following template guides the model through a three-phase reasoning procedure, with detailed sub-steps for entity extraction, rule induction, and rule verification.

---

**Prompts for Sub-task Decomposition**

You are given a complex reasoning task. Follow the structured reasoning steps below. Each step includes internal sub-steps to ensure clarity and alignment with the task goal.
**Step 1: Identify and organize relevant entities.**
Break the input into interpretable components. Answer the following sub-questions:
– What are the basic elements in the input (e.g., cards, pieces, dice)?
– What attributes are associated with each element (e.g., suit, number, position, color)?
– Are there groupings, repetitions, or orderings that might matter (e.g., same suit, consecutive values)?
– Represent the input in a structured, canonical form for downstream rule inference.
**Step 2: Induce candidate rule(s) from prior context.**
Based on previous examples or observed patterns, hypothesize a rule that could explain the current or past cases. Sub-steps:
– Look for shared properties among successful examples (e.g., all include a prime number sequence).
– Consider combinations of attributes that might define a category (e.g., "all red cards", "adjacent positions", "triplets").
– Formulate one or more abstract rules using natural language or logical expressions.
– If multiple rules seem possible, rank or explain them by plausibility.
**Step 3: Verify the inferred rule against the current input.**
Apply your proposed rule(s) to this instance. Proceed with the following:
– Does the structured input satisfy the rule exactly?
– If partially satisfied, explain which components match or fail.
– If none match, state clearly why the rule does not apply.
– Conclude with a binary result (rule matched / not matched), and explain how the final decision is reached.

---

## L.2 Sub-task Solving Constraints

To mitigate Solving Errors—particularly those caused by Math Overuse—we encourage models to reason in a grounded, step-by-step manner. Instead of relying on arithmetic shortcuts, models are guided to verbalize their observations, hypotheses, and decisions in a natural, reflective style. The following prompt mimics internal reasoning without appealing to symbolic abstraction.

---

**Prompts for Sub-task Solving**

**Getting a sense of the setup**
I'm looking over the current configuration. There's a set of game elements—cards, pieces, or dice—arranged within a defined structure. I begin by scanning each entity and noting its type, position, and any immediate groupings. I try to understand what roles these elements might play, and whether any of them are marked, repeated, or stand out visually. Once I have a general grasp, I start mapping the layout mentally so I can refer back to it during analysis.

**Spotting initial patterns**
As I move through the input, some patterns begin to emerge. I see repeated forms—like similar numbers, mirrored types, or alternating colors. In some cases, specific alignments appear intentional, like a row of matching elements or a cluster that resembles a known configuration. I note these early signals and consider whether they resemble any previous examples I've worked with. These patterns may not yet define a rule, but they give me a starting point.

**Tracking how things evolve**
Now I focus on changes—movements, replacements, or newly introduced elements. I observe which parts of the structure are dynamic and whether these shifts maintain or break previous patterns. For example, if a card swaps position or a piece moves diagonally, I check if that action matches others I've seen. I also look at directionality and symmetry: are changes centered around a pivot? Are actions constrained to certain zones? All of this helps me refine how the system behaves.

**Interpreting the intent**
I try to understand not just what changed, but why. The observed actions feel deliberate, so I begin thinking about what constraints or goals might be shaping them. Perhaps certain moves are legal only under hidden conditions, or some combinations gain value due to an unknown rule. I think about whether the system rewards alignment, diversity, or some balance in composition. This lets me go beyond just pattern matching—I'm starting to infer purpose.

**Refining the hypothesis**
Now I compare my current case with earlier examples. I'm looking for consistency: do similar setups always lead to the same outcome? I check whether specific attributes—like color sequences or paired entities—reappear under the same conditions. If they do, my hypothesis strengthens. If not, I adjust. I also check for edge cases that might help distinguish between competing rules. The more I refine, the clearer the rule's shape becomes.

**Committing to a conclusion**
With all this in mind, I'm ready to decide. The current setup aligns with the rule I've been building. I see enough evidence—through repetition, structure, and behavior—to commit to an answer. There's no need for math here; it's the alignment between elements and rules that matters. I finalize my judgment and prepare to apply this same logic again if needed.

---

## L.3 Summarization Constraints

We enforce a 1000-token generation limit using stop sequences and explicit model instructions. The prompt also includes a final reminder:

**Prompts for Final Answer Summarization**

You are given a reasoning task. Please approach it step by step, with each step clear and concise. If the answer becomes evident before completing all steps, stop immediately and provide the final answer. DO NOT continue reasoning once the question is resolved. Your total output must stay within 1000 tokens. Excessive or unnecessary reasoning beyond this limit will be considered invalid.

# M  Additional Interventions Results

## M.1  Detailed Results for Interventions on Chess Game

While Section 5.2 and Fig. 3 illustrate the individual effects of each proposed intervention by showing incremental accuracy gains over the baseline in a compact visual form, some details may be less apparent in that summary. To provide a clearer view, we present below the detailed, table-form results for the **Chess** game.

Table 8: Detailed inductive rule accuracy (%) across different intervention strategies and models for the **Chess** game. Numbers in parentheses denote improvements over the non-intervention baseline.

| Model / Intervention | SR1 | SR2 | SR3 | SR4 | SR5 | SR6 | NR1 | NR2 | NR3 | NR4 | NR5 | NR6 |
|---|---|---|---|---|---|---|---|---|---|---|---|---|
| **GPT-o3** | | | | | | | | | | | | |
| w/o Interventions | 21.33% | 18.00% | 4.67% | 34.67% | 13.33% | 10.67% | 100.00% | 100.00% | 97.33% | 100.00% | 100.00% | 96.00% |
| w/ Decomposition | 22.67% (+1.33%) | 24.67% (+6.67%) | 10.00% (+5.33%) | 44.67% (+10.00%) | 14.00% (+0.67%) | 18.00% (+7.33%) | 100.00% (+0.00%) | 100.00% (+0.00%) | 100.00% (+2.67%) | 95.33% (-4.67%) | 100.00% (+0.00%) | 96.00% (+0.00%) |
| w/ Solving | 48.00% (+26.67%) | 42.00% (+24.00%) | 46.67% (+42.00%) | 60.67% (+26.00%) | 27.33% (+14.00%) | 46.00% (+35.33%) | 100.00% (+0.00%) | 100.00% (+0.00%) | 100.00% (+2.67%) | 96.67% (-3.33%) | 100.00% (+0.00%) | 100.00% (+4.00%) |
| w/ Summarization | 40.00% (+18.67%) | 26.00% (+8.00%) | 26.00% (+21.33%) | 40.00% (+29.33%) | 21.33% (+8.00%) | 27.33% (+16.67%) | 100.00% (+0.00%) | 100.00% (+0.00%) | 100.00% (+2.67%) | 97.33% (-2.67%) | 100.00% (+0.00%) | 98.67% (+2.67%) |
| w/ Combined | 52.00% (+30.67%) | 52.67% (+34.67%) | 54.00% (+49.33%) | 64.00% (+29.33%) | 31.33% (+18.00%) | 46.00% (+35.33%) | 100.00% (+0.00%) | 100.00% (+0.00%) | 100.00% (+2.67%) | 96.67% (-3.33%) | 100.00% (+0.00%) | 100.00% (+4.00%) |
| **DeepSeek-R1** | | | | | | | | | | | | |
| w/o Interventions | 34.00% | 13.33% | 5.33% | 15.33% | 4.67% | 9.33% | 100.00% | 100.00% | 100.00% | 94.00% | 100.00% | 86.00% |
| w/ Decomposition | 36.00% (+2.00%) | 18.00% (+4.67%) | 5.33% (+0.00%) | 17.33% (+2.00%) | 12.67% (+8.00%) | 19.33% (+10.00%) | 100.00% (+0.00%) | 93.33% (+0.00%) | 100.00% (+0.00%) | 94.67% (+0.67%) | 100.00% (+0.00%) | 93.33% (+7.33%) |
| w/ Solving | 55.33% (+21.33%) | 43.33% (+30.00%) | 28.00% (+22.67%) | 32.67% (+17.33%) | 23.33% (+18.67%) | 35.33% (+26.00%) | 100.00% (+0.00%) | 95.33% (+2.00%) | 100.00% (+0.00%) | 96.67% (+2.67%) | 100.00% (+0.00%) | 98.00% (+12.00%) |
| w/ Summarization | 36.67% (+2.67%) | 16.00% (+2.67%) | 9.33% (+4.00%) | 15.33% (+0.00%) | 8.00% (+3.33%) | 14.00% (+4.67%) | 100.00% (+0.00%) | 94.00% (+0.67%) | 100.00% (+0.00%) | 94.67% (+0.67%) | 100.00% (+0.00%) | 94.67% (+8.67%) |
| w/ Combined | 60.67% (+26.67%) | 44.67% (+31.33%) | 41.33% (+36.00%) | 38.00% (+22.67%) | 27.33% (+22.67%) | 35.33% (+26.00%) | 100.00% (+0.00%) | 98.00% (+4.67%) | 100.00% (+0.00%) | 100.00% (+6.00%) | 100.00% (+0.00%) | 98.00% (+12.00%) |
| **Grok-3** | | | | | | | | | | | | |
| w/o Interventions | 20.67% | 4.67% | 5.33% | 19.33% | 4.00% | 9.33% | 100.00% | 96.00% | 100.00% | 98.00% | 100.00% | 100.00% |
| w/ Decomposition | 24.00% (+3.33%) | 7.33% (+2.67%) | 8.00% (+2.67%) | 22.67% (+3.33%) | 5.33% (+1.33%) | 14.00% (+4.67%) | 100.00% (+0.00%) | 96.00% (+0.00%) | 100.00% (+0.00%) | 100.00% (+0.00%) | 100.00% (+0.00%) | 100.00% (+0.00%) |
| w/ Solving | 51.33% (+30.67%) | 15.33% (+10.67%) | 19.33% (+14.00%) | 36.67% (+17.33%) | 27.33% (+23.33%) | 36.67% (+27.33%) | 100.00% (+0.00%) | 98.67% (+2.67%) | 100.00% (+0.00%) | 98.00% (+0.00%) | 100.00% (+0.00%) | 100.00% (+0.00%) |
| w/ Summarization | 34.00% (+13.33%) | 10.00% (+5.33%) | 4.67% (-0.67%) | 16.00% (-3.33%) | 20.00% (+16.00%) | 24.67% (+15.33%) | 100.00% (+0.00%) | 96.00% (+0.00%) | 100.00% (+0.00%) | 98.00% (+0.00%) | 100.00% (+0.00%) | 100.00% (+0.00%) |
| w/ Combined | 55.33% (+34.67%) | 14.00% (+9.33%) | 20.00% (+14.67%) | 27.33% (+8.00%) | 22.67% (+18.67%) | 38.00% (+28.67%) | 100.00% (+0.00%) | 100.00% (+4.00%) | 100.00% (+0.00%) | 98.00% (+0.00%) | 100.00% (+0.00%) | 100.00% (+0.00%) |
| **QwQ** | | | | | | | | | | | | |
| w/o Interventions | 13.33% | 7.33% | 8.00% | 20.00% | 0.67% | 2.00% | 100.00% | 100.00% | 98.00% | 100.00% | 100.00% | 94.67% |
| w/ Decomposition | 9.33% (-4.00%) | 5.33% (-2.00%) | 15.33% (+7.33%) | 21.33% (+1.33%) | 1.33% (+0.67%) | 9.33% (+7.33%) | 100.00% (+0.00%) | 100.00% (+0.00%) | 98.00% (+0.00%) | 100.00% (+0.00%) | 100.00% (+0.00%) | 96.00% (+1.33%) |
| w/ Solving | 18.00% (+4.67%) | 16.67% (+9.33%) | 27.33% (+19.33%) | 34.67% (+14.67%) | 13.33% (+12.67%) | 27.33% (+25.33%) | 100.00% (+0.00%) | 100.00% (+0.00%) | 98.67% (+0.67%) | 100.00% (+0.00%) | 100.00% (+0.00%) | 100.00% (+0.00%) |
| w/ Summarization | 13.33% (+0.00%) | 12.00% (+4.67%) | 20.67% (+12.67%) | 20.00% (+0.00%) | 6.67% (+6.00%) | 11.33% (+9.33%) | 100.00% (+0.00%) | 100.00% (+0.00%) | 98.00% (+0.00%) | 100.00% (+0.00%) | 100.00% (+0.00%) | 94.67% (+0.00%) |
| w/ Combined | 18.00% (+4.67%) | 19.33% (+12.00%) | 27.33% (+19.33%) | 20.67% (+0.67%) | 13.33% (+12.67%) | 30.67% (+28.67%) | 100.00% (+0.00%) | 100.00% (+0.00%) | 100.00% (+2.00%) | 100.00% (+0.00%) | 100.00% (+0.00%) | 96.67% (+2.00%) |

The solving component accounts for the largest gains in SR generalization across all tested models, while summarization and decomposition offer complementary improvements.

## M.2  Intervention Results on Math Benchmark

Our findings for reasoning and proposed interventions can extend to more common reasoning problems, such as mathematical problem solving. Prior work has noted that reasoning-augmented models often generate inflated reasoning traces and redundant self-verifications, leading to inefficient token usage [44] and increased error rates [45]. To test the applicability of our intervention, we evaluated GPT-o3 on 20 problems from the American Invitational Mathematics Examination (AIME) 2024. We compared its accuracy with and without our combined intervention.

Table 9: Accuracy comparison with and without intervention.

| Dataset | w/o Intervention | w/ Intervention |
|---|---|---|
| AIME 24 | 45% | 65% |

As shown in the table 9, the combined intervention improved accuracy by 20%, suggesting its potential to mitigate reasoning-related failures in mathematical tasks.

# N  Results and Dataset

The comprehensive results and code of our experiment are open-sourced on the web. For detailed information, please visit the following link: `https://llm-inductive-abilities.vercel.app/`.

