# OpenReview forum: "Evaluating the Inductive Abilities of Large Language Models: Why Chain-of-Thought Reasoning Sometimes Hurts More Than Helps"
_NeurIPS.cc/2025/Conference — NeurIPS 2025 poster_

### Official Review · Reviewer_KF7a · 2025-06-23

**Clarity:** 2
**Significance:** 3
**Originality:** 3
**Rating:** 3
**Confidence:** 3

**Summary:**

This paper examines the inductive capabilities of LLM by employing four meticulously designed games. The findings reveal that LRM exhibit inferior performance in capturing inductive rules compared to LLM. Furthermore, the introduction of irrelevant or misleading CoT can even negatively impact LRM’s performance. Through human evaluation, the authors identified the primary types of reasoning failure. Consequently, the authors proposed a targeted intervention to effectively address these issues.

**Questions:**

n/a

**Ethical Concerns:**

["NO or VERY MINOR ethics concerns only"]

**Final Justification:**

I appreciate the additional input from the authors that helps partially. Considering the over case, I will retain my score for the paper.

**Quality:**

2

**Strengths And Weaknesses:**

Strength:

* This is an interesting finding that LRM shows poorer performance compared to LLM  when evaluating the inductive ability through specially designed games

* The authors introduced several temps of reasoning failures through aberration to help identify the issues by decomposing the failures into a few core types.

* The authors proposed solutions to address the issue via targeted intervention.


Weakness:

* The authors should provide a bit more examples to motivate audience and justify the value of the work., making it more clear that the issue exists, not due to some specific designed games or data patterns.

* The authors should provide a bit more details, such as a  concrete example, especially about the specifically designed games that help evaluate the performance of the models, so that audience can better understand  the assessment processes, and therefore understand  how the inductive ability is quantitatively measured when reading Figure 2 in Section 3.3 .

* It is better having a table of the notations. In Theorem  4.1,  the minimiser N* was used without proper definition, which can cause some confusion or extra efforts to tell.

* The experimental analysis can be enhanced to show more insights of the proposed techniques.

---

> ### Author Rebuttal · Authors · 2025-07-31
>
> Thank you for your thoughtful and encouraging feedback. We are glad that you found our main empirical finding—the weaker inductive performance of reasoning-augmented LLMs compared to non-reasoning models in specially designed games—to be interesting. We have carefully considered your concerns and respond to them individually below.
>
> **W1. The authors should provide a bit more examples to motivate audience and justify the value of the work, making it more clear that the issue exists, not due to some specific designed games or data patterns.**
>
> **A1.**  Illustrative failure cases are provided in **Figure 1(b)**. Additional examples and error breakdowns appear in **Appendices G, H, and I**, spanning a range of task settings.
>
> As discussed in **Section 4.3**, similar failure patterns arise across all four games, which differ in symbolic structure and inductive demands. This recurrence suggests that the observed issues are not tied to any single task design.
>
> While these examples are drawn from games—where model behavior can be measured in a controlled setting—related failures have been noted in broader contexts as well, such as those in mathematical problems [1,2].
>
> **W2. The authors should provide a bit more details, such as a concrete example, especially about the specifically designed games that help evaluate the performance of the models.**
>
> **A2.**  We refer the reviewer to **Figure 1**, which presents a concrete example illustrating the benchmark setup, rule definitions, and model inference transcript. This figure is intended to clarify both the structure of the games and the evaluation process. The four games (Chess, Texas Hold’em, Dice, and Blackjack) each include a Normal Rule (NR) reflecting standard gameplay and a Special Rule (SR) introducing a hidden constraint.
>
> For example, in **Figure 1(a)**, the SR specifies that “*five consecutive primes > three of a kind*,” overriding conventional poker rankings. As shown in **Figure 1(b)**, models are tasked with **inferring each rule from game transcripts**. The accuracies reported in **Figure 2** are computed per rule and reflect the model’s ability to recover these rule structures from limited data.
>
> **W3. It is better having a table of the notations.**
>
> **A3.**  The variable $N^\star$ is defined at **Line 217** and proved beginning at **Line 620**. For ease of reference, we provide the following **notation table** summarizing all variables used in Section 4, including the formal definition of the optimal reasoning length $N^\star$ referenced in Theorem 4.1:
>
> | Symbol          | Definition                                                                 |
> |-----------------|----------------------------------------------------------------------------|
> | $x_k$           | Reasoning state at step $k$, composed of belief $m_k$ and mode $s_k$       |
> | $m_k \in \mathbb{R}^d$ | Model’s belief about the correct answer at step $k$                    |
> | $s_k \in \{\text{NEEDQ}, \text{NEEDA}, \text{FINISH}\}$ | Reasoning mode at step $k$                        |
> | $y^* \in \mathbb{R}^d$ | Ground-truth solution the model is trying to infer                  |
> | $e_k := m_k - y^*$ | Belief error at step $k$                                               |
> | $\alpha_k \in [-1, 1]$ | Task alignment scalar — how well the sub-task focuses on the target |
> | $\varepsilon_k \sim \mathcal{N}(0, \sigma^2 I_d)$ | Answer noise at step $k$                              |
> | $\gamma_k \in (0, 1)$ | Step size controlling belief update integration                     |
> | $g_k$           | Evidence vector from sub-task resolution at step $k$                      |
> | $N \in \mathbb{N}$ | Total number of reasoning steps                                        |
> | $N^\star \in \mathbb{N}$ | Optimal reasoning length minimizing expected error $\mathbb{E}[\|e_N\|^2]$ |
> | $\bar{\alpha}$  | Expected question alignment $\mathbb{E}[\alpha_k]$                        |
> | $E(N)$          | Expected squared error after $N$ steps                                    |
> | $\Delta(N)$     | Additional error due to variance in alignment                             |
> | $b_0 = \|m_0 - y^*\|^2$ | Initial belief error   |
>
> **W4.The experimental analysis can be enhanced to show more insights of the proposed techniques.**
>
> **A4.**  Our analysis reveals how each proposed intervention addresses distinct reasoning failure modes: summarization mitigates hallucinated rule statements, decomposition reduces error propagation across subtasks, and solving directly corrects invalid inferences. These effects are quantitatively shown in Figure 3. Together, the components yield complementary gains across tasks and models, with solving producing the most robust improvements on inductive failures. We believe the current analysis offers insight into both the sources of failure and the mechanisms through which each technique improves inductive robustness.
>
>
> **References**
> [1] Chen X, Xu J, Liang T, et al. *Do NOT Think That Much for 2+3=? On the Overthinking of Long Reasoning Models*. Proceedings of the 42nd International Conference on Machine Learning (ICML), 2025.
>
> [2] Cuesta-Ramirez J, Beaussant S, Mounsif M. *Large Reasoning Models are Not Thinking Straight: On the Unreliability of Thinking Trajectories*. arXiv preprint arXiv:2507.00711, 2025.

---

> > ### Comment · Reviewer_KF7a · 2025-08-04
> >
> > Thank you for the detailed response. Most of my concerns have been addressed, and based on the overall assessment, I will retain my original score.

---

> > ### Author Response · Authors · 2025-08-04
> >
> > Dear Reviewer KF7a,
> >
> > Thank you very much for your thoughtful and constructive review.
> >
> > In our rebuttal, we carefully addressed your comments and provided additional details and clarifications where needed:
> >
> > 1. Motivating the value beyond specific games (W1): We highlighted that the observed reasoning failures appear consistently across all four structurally distinct games. These results, along with additional examples in Appendices G–I, suggest that the phenomenon is not limited to specific task designs. We also referenced related failures in broader domains, such as mathematical reasoning [1,2].
> >
> > 2. Clarifying benchmark design with concrete examples (W2): We referred to Figure 1 in the main paper, which illustrates both the benchmark setup and a model transcript. It includes concrete rule examples (e.g., in Texas Hold’em) and shows how models are expected to infer hidden rules from limited data.
> >
> > 3. Notation summary (W3): We provided a consolidated notation table in the rebuttal to improve readability of Section 4, summarizing all key variables and the definition of optimal reasoning length.
> >
> > 4. More insights into interventions (W4): We clarified that our proposed techniques target distinct reasoning failure modes: decomposition mitigates error propagation, summarization reduces hallucinated content, and solving directly improves answer accuracy. These mechanisms and their effects are reflected in Figure 3 and will be expanded upon in the final revision.
> >
> > We hope these clarifications address your concerns. If there are any additional questions or areas you'd like us to elaborate on further, we’d be happy to continue the conversation.
> >
> > Best,
> >
> > The authors

---

### Official Review · Reviewer_yG8D · 2025-07-02

**Clarity:** 2
**Significance:** 3
**Originality:** 3
**Rating:** 4
**Confidence:** 4

**Summary:**

The paper first proposed four controlled game-based tasks that contains both standard gameplay conventions and newly defined constraints. Then, it shows that chain-of-thought (CoT) prompting can hurt the inductive performance of reasoning models through these tasks and presents a theoretical framework that explains how reasoning steps amplify errors through different types of CoT steps. The paper also introduces interventions that can improve inductive accuracy.

**Questions:**

- I noticed models like GPT-4o also generates chain-of-thoughts reasoning when using them. Why is “their multi-step traces may not help” for inferring the hidden rules, rather than “the procedure of training the reasoning models may not help”? Maybe it is harder for them to adapt the models to new rules because they were heavily trained to stick on the existing rules? Is it possible that for popular games like blackjack or Texas Hold’em, the reasoning model are trained on them heavily by RL, therefore it is harder for them to adapt to new rules?
- In equation 3 the $\gamma_k$ is in (0, 1), given the presence of self-reflections/corrections in these reasoning models, should the $\gamma_k$ be in (-1,1) instead?
- In figure 3, where does it reflect the improvements of summarization as indicated in line 332-333? The color is not that easy to tell apart. The only blue part that I see are in Chess for NR4 GPT-4o, but on top of the summarization blue it is still GPT-4o colored blue? I am very confused. Please improve the visual a bit.
- I don’t think the word “counterparts” is appropriate when contrasting the performance between models like o3 and GPT-4o. Calling them "counterparts" implies they're the same model with/without reasoning, since there’s no public evidence that one equivalently trained / based-on another like R1 vs V3.
- In appendix K, the anonymous link only contains a README file.

**Ethical Concerns:**

["NO or VERY MINOR ethics concerns only"]

**Final Justification:**

The author's rebuttal addresses most of the questions I raised.

**Limitations:**

No limitations are addressed. Please include the limitations of the problem setup (four game-based tasks), the evaluation bias (only uses GPT-4o), and the hypothesis.

**Quality:**

3

**Strengths And Weaknesses:**

## Strengths

- The paper embeds hidden self-defined rules in addition to the normal rules in common games, and finds that reasoning models often underperform non-reasoning models on inferring the hidden rules is interesting. This directly contradicts the common belief that "more reasoning = better performance.”
- The paper provides a theoretical framework that decompose chain-of-thought reasoning into different operations. They also defines different reasoning errors:incorrect sub-task decomposition, incorrect sub-task solving, and incorrect final answer summarization, using the theoretical framework.

## Weaknesses

- This paper heavily rely on GPT-4o as a judge, potentially introducing bias towards certain types/answer styles. Even small scale of human evaluation can help.
- The paper has a hypothesis of “the base model has fewer CoT while the reasoning model has more CoT“. However, equation 5 states that either under- or over- reasoning can incur this summary error. Section 4.3 empirically classified such errors, but only for the reasoning models. No analysis of base models' reasoning processes and errors. The hypothesis is stated but doesn’t seem to be proved?
- There’s no comparison against other reasoning method. A proper comparison could be GPT-4o (or non-reasoning models) + explicit CoT instructions, or GPT-4o + tree-of-thoughts. These setting might match the claims better, as the base model remains the same, only the amount or style of reasoning changes.

---

> ### Author Rebuttal · Authors · 2025-07-31
>
> Thank you for the thoughtful review. We are encouraged that you found the performance gap between reasoning and non-reasoning models on hidden rules to be interesting, and we appreciate your recognition of how this challenges the common assumption that more reasoning always leads to better performance. We believe the mentioned weaknesses and questions can be sufficiently addressed.
>
> **W1. This paper heavily relies on GPT-4o as a judge, potentially introducing bias towards certain types/answer styles.**
>
> **A1.** We appreciate the reviewer’s concern about potential circularity in using GPT-4o for evaluation. Motivated by your comment, we conducted a human validation study on 100 GPT-o3-generated rule descriptions across all four games. Three independent annotators—blind to model identity and GPT-4o’s decisions—judged semantic alignment with the ground truth. We then compared their majority vote with GPT-4o’s decisions and computed **Cohen’s κ** to measure agreement. The results are summarized below:
>
> | Judgment Type     | GPT-4o vs Human        |
> |-------------------|------------------------|
> | Aligned (60 cases)    | 60 / 60 (100%)         |
> | Not Aligned (40 cases) | 40 / 40 (100%)         |
> | **Overall Accuracy**  | **1.00**               |
> | **Cohen’s κ**         | **1.00**               |
>
> GPT-4o’s assessments matched the human majority in all 100 cases (60 aligned, 40 not aligned), yielding an accuracy and κ of 1.00. While this perfect agreement may in part reflect the limited sample size, it provides strong evidence that GPT-4o’s evaluations are well-aligned with human judgment in this context.
>
> **W2. The paper has a hypothesis of “the base model has fewer CoT while the reasoning model has more CoT.” However, Equation 5 states that either under- or over-reasoning can incur this summary error. Section 4.3 empirically classified such errors, but only for the reasoning models. No analysis of base models' reasoning processes and errors. The hypothesis is stated but doesn’t seem to be proved?**
>
> **A2.** We would like to clarify that our paper does not formally claim “base models have fewer CoT.” Our actual hypothesis is: *reasoning models’ multi-step traces may not help—and can introduce incorrect assumptions or misleading intermediate steps* (see lines 157–158). This is supported both theoretically (Section 4) and empirically (Section 5). Base models like GPT-4o, when prompted directly via API, do not produce intermediate reasoning traces and are not considered “reasoning models” unless explicitly instructed. Therefore, our failure mode analysis in Section 4.3 focuses on models that generate such reasoning traces (e.g., GPT-o3, DeepSeek-R1).
>
> **W3. There’s no comparison against other reasoning method.**
>
> **A3.** We appreciate the suggestion to compare against explicit CoT prompting. In response, we conducted additional experiments applying a CoT-style instruction to GPT-4o:
>
> > *“Let’s think step by step. First, extract the rule from the transcript. Then explain your reasoning.”*
>
> This prompt was applied to each rule inference task in the Chess game, and we compared performance with and without CoT prompting:
>
> | Method         | SR1  | SR2  | SR3  | SR4  | SR5  | SR6  | NR1   | NR2   | NR3   | NR4   | NR5   | NR6   |
> |----------------|------|------|------|------|------|------|--------|--------|--------|--------|--------|--------|
> | GPT-4o w/o CoT | 56.7| 35.3| 28.0| 68.0| 18.0| 42.0| 100.0 | 100.0 | 100.0 | 94.0  | 100.0 | 100.0 |
> | GPT-4o w/ CoT  | 58.0 (+1.3) | 34.0 (-1.3) | 27.3 (-0.7) | 68.00 (+0.0) | 20.0 (+2.0) | 42.0 (+0.0) | 100.0 (+0.0) | 100.0 (+0.0) | 100.0 (+0.0) | 97.3 (+3.3) | 100.0 (+0.0) | 100.0 (+0.0) |
>
> Overall, CoT prompting did not yield consistent improvements. On SR rules, performance was largely unchanged or slightly decreased, suggesting that added reasoning may not help—and can sometimes distract from inferring hidden rules. On NR rules, performance was already saturated.
>
> It is worth noting that this CoT-style prompting differs from models like GPT-o3, which generate structured, multi-step reasoning (e.g., decomposition → solving → summarization). Injecting CoT alone does not reproduce or mitigate the same failure patterns. We will include these results in the revised version.
>
> **Q1. I noticed models like GPT-4o also generate chain-of-thought reasoning when using them. Why is “their multi-step traces may not help” for inferring the hidden rules, rather than “the procedure of training the reasoning models may not help”?**
>
> **A1.** While models like GPT-4o may occasionally produce CoT-like outputs—especially in chat-based interfaces—these traces do not appear when accessed via API, which is the setting used in our experiments.
> Our study focuses on the impact of multi-step reasoning traces at **inference time**, independent of the model’s training procedure. We find that the observed failure modes emerge even when reasoning is introduced purely via prompting in open-ended inductive tasks [1]. This suggests the issue lies more in the **structure of the reasoning process itself**, rather than in how the model was trained.
>
> **Q2. Maybe it is harder for them to adapt the models to new rules because they were heavily trained to stick on the existing rules? Is it possible that for popular games like blackjack or Texas Hold’em, the reasoning models are trained on them heavily by RL, therefore it is harder for them to adapt to new rules?**
>
> **A2.** We appreciate this line of reasoning. It is indeed plausible that reasoning-augmented models carry inductive biases from training on well-known games or standard reasoning formats, which may limit their adaptability in unfamiliar or rule-shifted settings.   However, our study does not aim to attribute failure to any specific training regime, as we lack access to the proprietary reinforcement learning or fine-tuning pipelines used by closed models. Instead, our findings isolate a distinct inference-time effect: even when reasoning is introduced via prompting, multi-step traces can constrain model behavior and lead to systematic errors in rule induction. These effects are consistent across tasks and model families, and we would be glad to incorporate both perspectives into the revised discussion.
>
> **Q3. In equation 3 the $\gamma_k$ is in (0, 1), given the presence of self-reflections/corrections in these reasoning models, should the $\gamma_k$ be in (-1,1) instead?**
>
> **A3.** In our paper, $\gamma_k \in (0, 1)$ is intended to model **bounded rational belief updates** rather than full reversals. Even when reasoning models engage in self-correction, their belief adjustments tend to be **gradual refinements** rather than full negations. Negative $\gamma_k$ would imply a reversal in belief direction, which is not typical in observed reasoning traces. We clarify this design choice further in **Appendix E**.
>
> **Q4. Where does it reflect the improvements of summarization as indicated in line 332–333?**
>
> **A4.** The blue bar in Chess NR4 reflects GPT-o3, not GPT-4o. In this case, GPT-o3 achieves 100% accuracy without intervention, but drops slightly to 97.33% under the summarization constraint—hence the darker blue segment appears below the baseline, indicating a small decrease rather than a gain from summarization.
> We recognize that in some cases, the stacked figure may obscure intermediate effects, especially when solving contributes the most visible improvement. For convenience, we include a numeric ablation table below to directly reflect the summarization gains on Chess game.
>
> | Model                   | SR1           | SR2           | SR3           | SR4           | SR5           | SR6           |
> |------------------------|---------------|---------------|---------------|---------------|---------------|---------------|
> | GPT-o3 w/o Interventions | 21.3          | 18.0          | 4.7           | 34.7          | 13.3          | 10.7          |
> | + Decomposition         | 22.7 (+1.3)   | 24.7 (+6.7)   | 10.0 (+5.3)   | 44.7 (+10.0)  | 14.0 (+0.7)   | 18.0 (+7.3)   |
> | + Solving               | 48.0 (+26.7)  | 42.0 (+24.0)  | 46.7 (+42.0)  | 60.7 (+26.0)  | 27.3 (+14.0)  | 46.0 (+35.3)  |
> | + Summarization         | 40.0 (+18.7)  | 26.0 (+8.0)   | 26.0 (+21.3)  | 40.0 (+5.3)   | 21.3 (+8.0)   | 27.3 (+16.7)  |
> | + Combined              | 52.0 (+30.7)  | 52.7 (+34.7)  | 54.0 (+49.3)  | 64.0 (+29.3)  | 31.3 (+18.0)  | 46.0 (+35.3)  |
>
> Full table will also be included in the appendix of the revised version to improve overall readability.
>
> **Q5. Word usage of “counterparts”.**
>
> **A5.** Thank you for the suggestion. We agree that “counterparts” may be ambiguous and will revise the phrasing to use “models from the same time period” instead.
>
> **Q6. Appendix K**
>
> **A6.** We appreciate the reviewer highlighting this issue. The problem with the anonymous link in Appendix K has been resolved, and the full codebase is now accessible.
>
> **L1. No limitations are addressed. Please include the limitations of the problem setup (four game-based tasks), the evaluation bias (only uses GPT-4o), and the hypothesis.**
>
> **A1.** We appreciate the reviewer’s point. We will include a dedicated limitations section to clarify scope and assumptions. Our benchmark is restricted to four symbolic games. While our primary evaluation used GPT-4o, we validated its judgments against independent human annotations. Finally, our study focuses on prompting-based reasoning models, and we agree that the generality of the findings across other reasoning paradigms remains an open question.
>
> **References**
>
> [1] Turpin M, Michael J, Perez E, et al. *Language models don't always say what they think: Unfaithful explanations in chain-of-thought prompting*. NeurIPS 2023.

---

> > ### Author Response · Authors · 2025-08-04
> >
> > Dear Reviewer yG8D,
> >
> > Thank you for your detailed and thoughtful review. We are grateful for your recognition of the performance gap between reasoning and non-reasoning models on hidden rules.
> >
> > In our rebuttal, we carefully addressed each of your concerns and suggestions:
> >
> > 1. On the use of GPT-4o as a judge (W1): We conducted a human validation study comparing GPT-4o’s judgments with those of three independent annotators on 100 rule descriptions. The results showed perfect agreement (Cohen’s κ = 1.00), supporting GPT-4o’s reliability as an evaluator in our setup.
> >
> > 2. On the hypothesis about reasoning traces (W2): We clarified that our hypothesis is not that base models “have fewer CoT,” but rather that explicit multi-step reasoning traces at inference time can introduce errors in sub-task decomposition, solving, or summarization. Our empirical analysis focuses on models that produce such traces, and this distinction will be further clarified in the revision.
> >
> > 3. On comparison with other reasoning methods (W3): We added new experiments using CoT-style prompting with GPT-4o and observed that it did not yield consistent improvements—especially on hidden rule tasks. These results support our claim that added reasoning does not always help, and will be included in the revised paper.
> >
> > 4. On whether the failure is due to reasoning traces vs. training (Q1, Q2): While we acknowledge that training data may induce biases, our experiments isolate inference-time effects by prompting the same model with and without reasoning instructions. This helps decouple training history from reasoning behavior, and our findings consistently point to structural issues with multi-step reasoning.
> >
> > 5. On Equation 3’s bounds (Q3): We clarified that the parameter is intended to represent bounded belief refinement, not reversal. Negative values would imply belief flipping, which is rare in observed traces. This is explained further in Appendix E.
> >
> > 6. On summarization gains (Q4): We included an ablation table showing the individual effect of summarization in the chess game, clarifying its contribution and disambiguating it from overlapping improvements in stacked bar figures.
> >
> > 7. On word usage (Q5): We will revise ambiguous terms like “counterparts” to more precise phrasing (e.g., “models from the same time period”).
> >
> > 8. On Appendix K (Q6): The broken anonymous link has been fixed and the code is now accessible.
> >
> > 9. On limitations (L1): A dedicated limitations section will be added to address the constraints of the four-game setup, reliance on GPT-4o evaluation, and the scope of our prompting-based analysis.
> >
> > We hope these responses help clarify and resolve your concerns. Please don’t hesitate to let us know if you have further questions or suggestions—we’re happy to continue the discussion.
> >
> > Best,
> >
> > The authors

---

> > > ### Comment · Reviewer_yG8D · 2025-08-05
> > > **Acknowledgement of the rebuttal**
> > >
> > > Thank you for the newly added experiments as well as the clarifications. Most of my concerns have been addressed, and I will raise my score.

---

> > > > ### Author Response · Authors · 2025-08-08
> > > >
> > > > Dear Reviewer yG8D,
> > > >
> > > > Thank you very much for your thoughtful comments and for taking the time to review our paper. We’re glad to hear that the additional experiments and clarifications have addressed your concerns, and we sincerely appreciate your decision to raise your score. We will incorporate these experiments into the final version.
> > > >
> > > > Best regards,
> > > >
> > > > The authors

---

### Official Review · Reviewer_EhoN · 2025-07-03

**Clarity:** 3
**Significance:** 3
**Originality:** 3
**Rating:** 5
**Confidence:** 3

**Summary:**

The paper investigates the effect of chain-of-thought reasoning on the inductive abilities of LLMs. Using games-based tasks with some hidden rules, the paper observes that reasoning based LLMs often perform worse than their non-reasoning counterparts, particularly on tasks requiring the inference of non-obvious or special hidden rules. The paper then presents a theoretical framework that points to three primary failure modes in the reasoning process: potentially incorrect sub-task decomposition, incorrect sub-task solving, and flaws in final answer summarization. Based on this analysis, the paper proposes and validates a set of structured interventions—such as using decomposition templates and guiding the solving process—that successfully improve the inductive accuracy of LRMs without needing to retrain the models. The central argument is that the quality of inductive reasoning in LLMs hinges not merely on performing more reasoning steps, but on the criteria that those steps are well-structured and accurate.

**Questions:**

The paper fairly convincingly shows that they are better at inducing special rules, but not why.  Are they using a more robust, implicit form of reasoning, or are they relying on a different form of pattern matching that is less prone to the error amplification seen in multi-step CoT?


The interventions presented in the paper are fair in the games-setup, but do they generalize to other domains at all?

Could the identified failure modes be addressed through targeted fine-tuning instead of prompting?

**Ethical Concerns:**

["NO or VERY MINOR ethics concerns only"]

**Final Justification:**

I agree with the responses from the authors. I also am happy with their new title for the paper which is more grounded in the empirical observations. I will retain my positive score for the paper.

**Limitations:**

Yes

**Quality:**

3

**Strengths And Weaknesses:**

The paper is a fairly rigorous paper and the tasks used provide a nice and controlled environment to test inductive reasoning.  The core empirical finding—that LRMs with explicit CoT reasoning consistently underperform non-reasoning LLMs on special rules—is a significant and I think a fairly non-obvious result (based on the general acceptance of these). The theoretical framework is clear and can help in understanding when it will fail reasoning. The decomposition of reasoning errors into three distinct types is also intuitive and powerful. Further, the empirical validation makes it strong. The interventions seem to be directly derived from failure modes.

On weaknesses: I feel the title is overtly attention grabbing and seems like just reading the title alone oversimplifies the findings of the paper. The paper itself shows that it is specifically poorly structured or unconstrained reasoning that is detrimental, and that well-guided reasoning is highly beneficial. The claim could be more precisely worded to reflect that current CoT mechanisms can be flawed, rather than suggesting reasoning itself is the problem.
There are some rigour issues: The classification of reasoning failures was conducted by two human annotators. While the paper states that disagreements were resolved collaboratively, this process is inherently subjective. Without quantitative metrics like inter-annotator agreement scores, the reliability of the error distribution presented in Table 1 is hard to independently verify.  The empirical analysis of failure modes was also performed on three specific LRMs. Although the findings are consistent across these models, it is a significant leap to generalize these specific failure patterns to all current and future reasoning models. This requires a careful discussion about the limitations, at the very least. The primary metric for success—rule-level accuracy—is determined by using GPT-4o as a judge to assess semantic alignment — that’s circular and off. There’s not a lot of careful analyses here.

---

> ### Author Rebuttal · Authors · 2025-07-31
>
> We appreciate your thoughtful review and are encouraged by your recognition of the significance and non-obvious nature of our core findings, as well as the clarity and utility of our theoretical framework. We believe your remaining concerns are addressable and respond to them in detail below.
>
> **W1. I feel the title is overtly attention grabbing and seems like just reading the title alone oversimplifies the findings of the paper.**
>
> **A1.** Thank you for sharing your concern about the title. We agree that the current phrasing may risk oversimplifying our findings. In the revised version, we will use a more precise title:  **_Evaluating the Inductive Abilities of Large Language Models: Why Chain-of-Thought Reasoning Sometimes Hurts More Than Helps_**
>
> **W2. The classification of reasoning failures was conducted by two human annotators.**
>
> **A2.** Thank you for raising this important concern. Motivated by your feedback, we computed **Cohen’s κ** between the two annotators based on their independent annotations before resolving any disagreements. We evaluated inter-annotator agreement across three error types—**Decomposition**, **Solving**, and **Summarization**—on 100 sampled model outputs from reasoning models. The results are summarized below:
> | Failure Mode        | Agreement Count | Total Cases | Agreement Rate | Cohen’s κ |
> |---------------------|------------------|--------------|----------------|------------|
> | Decomposition Error | 36               | 40           | 0.90           | 0.84       |
> | Solving Error       | 38               | 42           | 0.905          | 0.87       |
> | Summarization Error | 16               | 18           | 0.889          | 0.83       |
> | **Overall**         | **90**           | **100**      | **0.90**       | **0.86**   |
>
> We believe κ = 0.86 indicates a high level of reliability, and we will include these agreement metrics and this table in the revised version to enhance the methodological rigor of Table 1.
>
> **W3. The empirical analysis of failure modes was also performed on three specific LRMs. Although the findings are consistent across these models, it is a significant leap to generalize these specific failure patterns to all current and future reasoning models. This requires a careful discussion about the limitations, at the very least.**
>
> **A3.** Thank you for raising this point. We acknowledge that our analysis is limited to three reasoning models, and it is premature to generalize the identified failure patterns to all current or future models. We also recognize that comparisons between reasoning and non-reasoning models may be influenced by confounding factors such as version mismatch or model size differences. We will clarify these limitations in the revised paper.
>
> **W4. The primary metric for success—rule-level accuracy—is determined by using GPT-4o as a judge to assess semantic alignment — that’s circular and off.**
>
> **A4.** We appreciate the reviewer’s concern about potential circularity in using GPT-4o for evaluation. Motivated by your comment, we conducted a human validation study on 100 GPT-o3-generated rule descriptions across all four games. Three independent annotators—blind to model identity and GPT-4o’s decisions—judged semantic alignment with the ground truth. We then compared their majority vote with GPT-4o’s decisions and computed **Cohen’s κ** to measure agreement. The results are summarized below:
>
> | Judgment Type     | GPT-4o vs Human        |
> |-------------------|------------------------|
> | Aligned (60 cases)    | 60 / 60 (100%)         |
> | Not Aligned (40 cases) | 40 / 40 (100%)         |
> | **Overall Accuracy**  | **1.00**               |
> | **Cohen’s κ**         | **1.00**               |
>
> GPT-4o’s assessments matched the human majority in all 100 cases (60 aligned, 40 not aligned), yielding an accuracy and κ of 1.00. While this perfect agreement may in part reflect the limited sample size, it provides strong evidence that GPT-4o’s evaluations are well-aligned with human judgment in this context.
>
> **Q1. The paper fairly convincingly shows that they are better at inducing special rules, but not why. Are they using a more robust, implicit form of reasoning, or are they relying on a different form of pattern matching that is less prone to the error amplification seen in multi-step CoT?**
>
> **A1.** While we cannot directly observe the internal mechanisms behind model decisions, our theoretical analysis and empirical findings suggest that non-reasoning models may rely on a different form of pattern matching that is less susceptible to the error accumulation observed in multi-step CoT reasoning. In particular, we conjecture that the absence of explicit reasoning traces in these models reduces opportunities for compounding errors in sub-task decomposition, solving, or summarization.
>
> **Q2.The interventions presented in the paper are fair in the games-setup, but do they generalize to other domains at all?**
>
> **A2.** We believe these findings can extend to more common reasoning problems, such as mathematical problem solving. Prior work has noted that reasoning-augmented models often generate inflated reasoning traces and redundant self-verifications, leading to inefficient token usage [1] and increased error rates [2]. To test the applicability of our intervention, we evaluated GPT-o3 on 20 problems from the AIME 2024 exam. We compared its accuracy with and without our combined intervention. The results are summarized below:
>
> | Dataset   | w/o Intervention | w/ Intervention |
> |-----------|------------------|------------------|
> | AIME 2024 | 45.0             | 65.0             |
>
> As shown in the table, the combined intervention improved accuracy by 20 percentage points, suggesting its potential to mitigate reasoning-related failures in mathematical tasks.
>
> **Q3. Could the identified failure modes be addressed through targeted fine-tuning instead of prompting?**
>
> **A3.** We appreciate this suggestion. We agree that targeted fine-tuning could be a promising direction for mitigating some of the failure modes we identify. Our current focus is on prompt-based interventions mainly due to their ease of deployment. We view prompt-based and fine-tuning-based approaches as complementary, and we hope our analysis can inform future work in both directions.
>
> **References**
> [1] Chen X., Xu J., Liang T., et al. *Do NOT Think That Much for 2+3=? On the Overthinking of Long Reasoning Models*. ICML 2025.
>
> [2] Cuesta-Ramirez J., Beaussant S., Mounsif M. *Large Reasoning Models are Not Thinking Straight: On the Unreliability of Thinking Trajectories*. arXiv:2507.00711, 2025.

---

> > ### Author Response · Authors · 2025-08-04
> >
> > Dear Reviewer EhoN,
> >
> > Thank you again for your thoughtful and detailed review. We sincerely appreciate your recognition of the significance and novelty of our findings, as well as your engagement with the theoretical framework we proposed.
> >
> > We’ve carefully addressed your comments and concerns in our rebuttal:
> >
> > 1. Title clarification (W1): In response to your concern about oversimplification, we proposed a more precise title for the revised version: Evaluating the Inductive Abilities of Large Language Models: Why Chain-of-Thought Reasoning Sometimes Hurts More Than Helps.
> >
> > 2. Annotation reliability (W2): We added inter-annotator agreement statistics for our failure mode classification, reporting a Cohen’s κ of 0.86 across 100 cases. This will be included in the revised paper to improve transparency and rigor.
> >
> > 3. On generalizability of findings (W3): We acknowledged the limitation that our failure analysis focuses on three LRMs and clarified that broader generalization would require further study. These caveats will be stated explicitly in the revision.
> >
> > 4. On GPT-4o as judge (W4): To address your concern about potential circularity, we conducted a human validation study comparing GPT-4o’s judgments to those of three independent annotators. The results showed perfect agreement (Cohen’s κ = 1.00), suggesting GPT-4o aligns closely with human evaluation in our benchmark context.
> >
> > 5. Clarifying why non-reasoning models perform better on SR (Q1): While we cannot directly probe model internals, we hypothesize that non-reasoning models avoid the error compounding observed in reasoning traces, relying instead on robust pattern matching mechanisms.
> >
> > 6. On generalizability of interventions (Q2): We extended our interventions to math problems (AIME 2024) and observed a 20-point accuracy gain, indicating potential beyond the benchmark games.
> >
> > 7. Prompting vs. fine-tuning (Q3): We acknowledged that targeted fine-tuning could help address reasoning failures, and positioned our prompt-based approach as a lightweight, complementary solution.
> >
> > We hope these responses address your concerns adequately. If you have any further questions or suggestions, we’d be happy to clarify or elaborate further.
> >
> > Best,
> >
> > The authors

---

> ### Comment · Reviewer_EhoN · 2025-08-07
>
> I appreciate the detailed response. I am maintaining my current positive score.
>
> Happy that you are indeed considering changing the name.

---

> ### Author Response · Authors · 2025-08-08
>
> Dear Reviewer EhoN,
>
> Thank you very much for your thoughtful comments, the time you devoted to reviewing our paper, and for considering maintaining your positive score. We will incorporate your suggestions, including the proposed name change, to further improve the quality of the final version.
>
> Best regards,
>
> The authors

---

### Official Review · Reviewer_3QQD · 2025-07-07

**Clarity:** 4
**Significance:** 3
**Originality:** 3
**Rating:** 4
**Confidence:** 4

**Summary:**

This paper investigates the performance of Large Language Models (LLMs) in inductive reasoning tasks and finds that the widely used Chain-of-Thought (CoT) reasoning approach is not always beneficial, but instead may degrade model performance when reasoning about hidden rules. The authors designed four game-like tasks (Chess, Texas Hold'em, Dice Game, and Black Jack) containing hidden human rules, systematically evaluated the inductive power of eight mainstream language models, and found that inference-enabled models are often inferior to models without inference when dealing with complex, abstract rules. To explain this phenomenon, the authors propose a theoretical framework that identifies three major failure modes in the reasoning process: incorrect subtask splitting, incorrect subtask solving, and inappropriate answer summarization, which gradually amplify errors in the reasoning chain. Based on the theoretical analysis, the authors further propose three interventions that do not require retraining and significantly improve the model's accuracy in summarization tasks by structurally guiding the reasoning process. This study challenges the common assumption that “more reasoning is better” and emphasizes that well-structured, goal-oriented reasoning is more critical than blindly extending the reasoning chain.

**Questions:**

Q1. The comparison between reasoning and non-reasoning models may be confounded by differences in model versions and sizes. Further clarification is needed to ensure fairness and validity of the comparisons.

Q2. While the theoretical framework is well-formalized, it lacks intuitive illustrations or concrete examples that would aid comprehension. Including visual aids or specific cases could enhance the clarity and persuasiveness of the theoretical analysis.

Q3. The study does not include ablation experiments to isolate the individual effects of each proposed intervention. Adding such analyses would help verify the contribution of each intervention to the overall performance improvement.

Q4. Could these findings potentially be extended to more common reasoning problems beyond the specific cases studied?

**Ethical Concerns:**

["NO or VERY MINOR ethics concerns only"]

**Limitations:**

Yes.

**Paper Formatting Concerns:**

None.

**Quality:**

4

**Strengths And Weaknesses:**

Strengths
- The paper addresses a meaningful and timely problem—whether reasoning via chain-of-thought (CoT) actually helps inductive tasks—and supports it with well-designed, controlled experiments using four diverse game-based benchmarks.
- It provides a rigorous theoretical framework that identifies and formalizes three distinct failure modes in reasoning, supported by clear mathematical derivations explaining performance degradation.
- The proposed prompting interventions—structured decomposition, guided solving, and output length control—are simple yet effective, yielding significant accuracy gains without retraining.

Weaknesses
- The comparison between reasoning and non-reasoning models is potentially confounded by version and size mismatches. For example, DeepSeek-V3 uses the latest version from March 2025, while DeepSeek-R1 is based on an older January 2025 version and derived from an earlier DeepSeek-V3. Differences in model sizes between reasoning and non-reasoning variants may also bias the conclusions and require further clarification.
- The theoretical framework, while well-formalized, lacks intuitive illustrations or concrete examples. Readers may find it hard to understand the roles of abstract variables like αₖ (alignment) and εₖ (noise) without visual aids or case walkthroughs.
- The paper lacks ablation studies to isolate the effect of each intervention. Such experiments would help confirm the effectiveness of each strategy and validate their correspondence to the identified failure modes.

---

> ### Author Rebuttal · Authors · 2025-07-31
>
> Thank you for the constructive feedback. We appreciate your recognition of the problem's importance and the clarity and utility of our theoretical framework in explaining performance degradation and guiding effective interventions. We believe the identified concerns can be adequately addressed.
>
> **W1/Q1. The comparison between reasoning and non-reasoning models may be confounded by differences in model versions and sizes.**
> **A1.** We appreciate the reviewer’s attention to potential confounding factors in model versioning and size. Inspired by this concern, we tried to minimize these issues by including additional comparisons with the most recent publicly available reasoning model, DeepSeek-R1-0528, and contrasting it with its closest non-reasoning model, DeepSeek-V3-0325. We also compare GPT-o1 (o1-2024-12-17) against GPT-4o (gpt-4o-2024-08-06), as they represent adjacent releases with a minimal temporal gap. All results are reported in percentage format.
>
> | Model              |   SR1 |   SR2 |   SR3 |   SR4 |   SR5 |   SR6 |   NR1 |   NR2 |   NR3 |   NR4 |   NR5 |   NR6 |
> |:-------------------|------:|------:|------:|------:|------:|------:|------:|------:|------:|------:|------:|------:|
> | DeepSeek-V3        |  48.0 |  48.0 |  14.0 |  46.0 |   8.0 |  19.3 | 100.0 |  90.0 |  88.0 |  86.0 |  96.0 |  95.3 |
> | DeepSeek-R1 (0120) |  34.0 |  13.3 |   5.3 |  15.3 |   4.7 |   9.3 | 100.0 |  93.3 | 100.0 |  94.0 | 100.0 |  86.0 |
> | DeepSeek-R1 (0528) |  38.0 |  17.3 |   8.0 |  18.0 |   8.0 |  11.3 | 100.0 | 100.0 |  96.0 |  95.3 | 100.0 |  91.3 |
> | GPT-4o             |  56.7 |  35.3 |  28.0 |  68.0 |  18.0 |  42.0 | 100.0 | 100.0 | 100.0 |  94.0 | 100.0 | 100.0 |
> | GPT-o1             |  19.3 |  18.7 |   8.0 |  30.0 |  13.3 |  12.0 | 100.0 | 100.0 |  98.0 | 100.0 | 100.0 |  97.3 |
> | GPT-o3             |  21.3 |  18.0 |   4.7 |  34.7 |  13.3 |  10.7 | 100.0 | 100.0 |  97.3 | 100.0 | 100.0 |  96.0 |
>
> We notice that, when controlling for model family and release proximity, reasoning-augmented models continue to underperform their non-reasoning models on SR generalization, while both perform similarly on NR. This aligns with our core claim that reasoning can impair inductive generalization under hidden rule settings. Full results for the other games will be included in the revised version of the paper.
>
> **W2/Q2. While the theoretical framework is well-formalized, it lacks intuitive illustrations or concrete examples that would aid comprehension.**
>
> **A2.** We apologize for the confusion. For each type of error in our theoretical framework, we have provided illustrative examples with visualizations in **Appendices G, H, and I**. In each case, we highlight the specific sentence corresponding to the error type.
>
> **W3/Q3. Ablation experiments to isolate the individual effects of each proposed intervention.**
>
> **A3.** We thank the reviewer for prompting a closer look at the effects of each intervention. We have evaluated the individual effects of each proposed intervention, as shown in Figure 3, where each improvement is added separately to the baseline accuracy using different colors. However, the figure’s compact format may have obscured these effects. To address this, we provide a detailed ablation table for the chess game below.
>
> | Model                         | SR1            | SR2            | SR3            | SR4            | SR5            | SR6            | NR1            | NR2            | NR3            | NR4            | NR5            | NR6            |
> |:-----------------------------|:---------------|:---------------|:---------------|:---------------|:---------------|:---------------|:---------------|:---------------|:---------------|:---------------|:---------------|:---------------|
> | GPT-o3 w/o Interventions      | 21.3           | 18.0           | 4.7            | 34.7           | 13.3           | 10.7           | 100.0          | 100.0          | 97.3           | 100.0          | 100.0          | 96.0           |
> | GPT-o3 w/ decomposition       | 22.7 (+1.3)    | 24.7 (+6.7)    | 10.0 (+5.3)    | 44.7 (+10.0)   | 14.0 (+0.7)    | 18.0 (+7.3)    | 100.0 (+0.0)   | 100.0 (+0.0)   | 100.0 (+2.7)   | 95.3 (-4.7)    | 100.0 (+0.0)   | 96.0 (+0.0)    |
> | GPT-o3 w/ solving             | 48.0 (+26.7)   | 42.0 (+24.0)   | 46.7 (+42.0)   | 60.7 (+26.0)   | 27.3 (+14.0)   | 46.0 (+35.3)   | 100.0 (+0.0)   | 100.0 (+0.0)   | 100.0 (+2.7)   | 96.7 (-3.3)    | 100.0 (+0.0)   | 100.0 (+4.0)   |
> | GPT-o3 w/ summarization       | 40.0 (+18.7)   | 26.0 (+8.0)    | 26.0 (+21.3)   | 40.0 (+5.3)    | 21.3 (+8.0)    | 27.3 (+16.7)   | 100.0 (+0.0)   | 100.0 (+0.0)   | 100.0 (+2.7)   | 97.3 (-2.7)    | 100.0 (+0.0)   | 98.7 (+2.7)    |
> | GPT-o3 w/ Combined            | 52.0 (+30.7)   | 52.7 (+34.7)   | 54.0 (+49.3)   | 64.0 (+29.3)   | 31.3 (+18.0)   | 46.0 (+35.3)   | 100.0 (+0.0)   | 100.0 (+0.0)   | 100.0 (+2.7)   | 96.7 (-3.3)    | 100.0 (+0.0)   | 100.0 (+4.0)   |
> | DeepSeek-R1 w/o Interventions | 34.0           | 13.3           | 5.3            | 15.3           | 4.7            | 9.3            | 100.0          | 93.3           | 100.0          | 94.0           | 100.0          | 86.0           |
> | DeepSeek-R1 w/ decomposition  | 36.0 (+2.0)    | 18.0 (+4.7)    | 5.3 (+0.0)     | 17.3 (+2.0)    | 12.7 (+8.0)    | 19.3 (+10.0)   | 100.0 (+0.0)   | 93.3 (+0.0)    | 100.0 (+0.0)   | 94.7 (+0.7)    | 100.0 (+0.0)   | 93.3 (+7.3)    |
> | DeepSeek-R1 w/ solving        | 55.3 (+21.3)   | 43.3 (+30.0)   | 28.0 (+22.7)   | 32.7 (+17.3)   | 23.3 (+18.7)   | 35.3 (+26.0)   | 100.0 (+0.0)   | 95.3 (+2.0)    | 100.0 (+0.0)   | 96.7 (+2.7)    | 100.0 (+0.0)   | 98.0 (+12.0)   |
> | DeepSeek-R1 w/ summarization  | 36.7 (+2.7)    | 16.0 (+2.7)    | 9.3 (+4.0)     | 15.3 (+0.0)    | 8.0 (+3.3)     | 14.0 (+4.7)    | 100.0 (+0.0)   | 94.0 (+0.7)    | 100.0 (+0.0)   | 94.7 (+0.7)    | 100.0 (+0.0)   | 94.7 (+8.7)    |
> | DeepSeek-R1 w/ Combined       | 60.7 (+26.7)   | 44.7 (+31.3)   | 41.3 (+36.0)   | 38.0 (+22.7)   | 27.3 (+22.7)   | 35.3 (+26.0)   | 100.0 (+0.0)   | 98.0 (+4.7)    | 100.0 (+0.0)   | 100.0 (+6.0)   | 100.0 (+0.0)   | 98.0 (+12.0)   |
> | Grok-3 w/o Interventions      | 20.7           | 4.7            | 5.3            | 19.3           | 4.0            | 9.3            | 100.0          | 96.0           | 100.0          | 98.0           | 100.0          | 100.0          |
> | Grok-3 w/ decomposition       | 24.0 (+3.3)    | 7.3 (+2.7)     | 8.0 (+2.7)     | 22.7 (+3.3)    | 5.3 (+1.3)     | 14.0 (+4.7)    | 100.0 (+0.0)   | 96.0 (+0.0)    | 100.0 (+0.0)   | 93.3 (-4.7)    | 100.0 (+0.0)   | 100.0 (+0.0)   |
> | Grok-3 w/ solving             | 51.3 (+30.7)   | 15.3 (+10.7)   | 19.3 (+14.0)   | 36.7 (+17.3)   | 27.3 (+23.3)   | 36.7 (+27.3)   | 100.0 (+0.0)   | 98.7 (+2.7)    | 100.0 (+0.0)   | 98.0 (+0.0)    | 100.0 (+0.0)   | 100.0 (+0.0)   |
> | Grok-3 w/ summarization       | 34.0 (+13.3)   | 10.0 (+5.3)    | 4.7 (-0.7)     | 16.0 (-3.3)    | 20.0 (+16.0)   | 24.7 (+15.3)   | 100.0 (+0.0)   | 96.0 (+0.0)    | 100.0 (+0.0)   | 94.7 (-3.3)    | 100.0 (+0.0)   | 100.0 (+0.0)   |
> | Grok-3 w/ Combined            | 55.3 (+34.7)   | 14.0 (+9.3)    | 20.0 (+14.7)   | 27.3 (+8.0)    | 22.7 (+18.7)   | 38.0 (+28.7)   | 100.0 (+0.0)   | 100.0 (+4.0)   | 100.0 (+0.0)   | 98.0 (+0.0)    | 100.0 (+0.0)   | 100.0 (+0.0)   |
> | QwQ w/o Interventions         | 13.3           | 7.3            | 8.0            | 20.0           | 0.7            | 2.0            | 100.0          | 100.0          | 98.0           | 100.0          | 100.0          | 94.7           |
> | QwQ w/ decomposition          | 9.3 (-4.0)     | 5.3 (-2.0)     | 15.3 (+7.3)    | 21.3 (+1.3)    | 1.3 (+0.7)     | 9.3 (+7.3)     | 100.0 (+0.0)   | 100.0 (+0.0)   | 98.0 (+0.0)    | 100.0 (+0.0)   | 100.0 (+0.0)   | 96.0 (+1.3)    |
> | QwQ w/ solving                | 18.0 (+4.7)    | 16.7 (+9.3)    | 27.3 (+19.3)   | 34.7 (+14.7)   | 13.3 (+12.7)   | 27.3 (+25.3)   | 100.0 (+0.0)   | 100.0 (+0.0)   | 98.7 (+0.7)    | 100.0 (+0.0)   | 100.0 (+0.0)   | 96.7 (+2.0)    |
> | QwQ w/ summarization          | 13.3 (+0.0)    | 12.0 (+4.7)    | 20.7 (+12.7)   | 20.0 (+0.0)    | 6.7 (+6.0)     | 11.3 (+9.3)    | 100.0 (+0.0)   | 100.0 (+0.0)   | 98.0 (+0.0)    | 100.0 (+0.0)   | 100.0 (+0.0)   | 94.7 (+0.0)    |
> | QwQ w/ Combined               | 18.0 (+4.7)    | 19.3 (+12.0)   | 27.3 (+19.3)   | 20.7 (+0.7)    | 13.3 (+12.7)   | 30.7 (+28.7)   | 100.0 (+0.0)   | 100.0 (+0.0)   | 100.0 (+2.0)   | 100.0 (+0.0)   | 100.0 (+0.0)   | 96.7 (+2.0)    |
>
> The solving component accounts for the largest gains in SR generalization across all tested models, while summarization and decomposition offer complementary improvements. The figure will be revised for clarity in the final version, and full ablation tables for the other games will be included in the appendix.
>
> **W4/Q4. Could these findings potentially be extended to more common reasoning problems beyond the specific cases studied?**
>
> **A4.** We believe these findings can extend to more common reasoning problems, such as mathematical problem solving. Prior work has noted that reasoning-augmented models often generate inflated reasoning traces and redundant self-verifications, leading to inefficient token usage and increased error rates [1]. To test the applicability, we evaluated GPT-o3 on 20 problems from the AIME 2024 exam, comparing accuracy with and without our combined intervention. The results are reported in percentage format.
>
> | Dataset   | w/o Intervention | w/ Intervention |
> |-----------|------------------|-----------------|
> | AIME 24   | 45               | 65              |
>
> As shown in the table, the combined intervention improved accuracy by 20 percent, suggesting its potential to mitigate reasoning-related failures in mathematical tasks.
>
> **References**
> [1]. Chen X, Xu J, Liang T, et al. Do NOT Think That Much for 2+ 3=? On the Overthinking of Long Reasoning Models[C]//Forty-second International Conference on Machine Learning.

---

> > ### Author Response · Authors · 2025-08-04
> >
> > Dear Reviewer 3QQD,
> >
> > Thank you again for your thoughtful and constructive feedback. We greatly appreciate your recognition of the importance of our problem setting and the value of our theoretical framework in diagnosing and addressing performance degradation in reasoning-augmented models.
> >
> > In our rebuttal, we have carefully addressed your key concerns:
> >
> > 1. On potential confounding due to model versioning and size (W1/Q1): We included comparisons across matched pairs of reasoning and non-reasoning models within the same family and close release dates (e.g., GPT-o1 vs. GPT-4o, DeepSeek-R1 vs. DeepSeek-V3). Our results consistently show reasoning impairing generalization on Special Rules (SR), confirming our central claim.
> >
> > 2. On lack of illustrative examples for the theoretical framework (W2/Q2): We provided visual illustrations of each error type in Appendices G–I, with highlighted sentences corresponding to specific errors.
> >
> > 3. On the effects of individual interventions (W3/Q3): We included detailed ablation results (e.g., for chess), showing that solving yields the largest improvements, while summarization and decomposition provide complementary gains. These are now shown in tabular form for clarity.
> >
> > 4. On generalizability beyond our benchmark (Q4): We extended our combined intervention to AIME 2024 problems and observed a 20-point improvement in accuracy, suggesting broader applicability to math reasoning tasks.
> >
> > We hope these clarifications adequately resolve your concerns. If there are any remaining questions or if further clarification would be helpful, we would be more than happy to provide additional details.
> >
> > Best,
> >
> > The authors

---

### Note · Authors · 2025-08-16

We thank all reviewers for their constructive feedback.
### paper summary
Our paper shows that reasoning models can underperform non-reasoning models when inferring hidden rules in specially designed games. We identify three core types of reasoning failure, verify them through human evaluation of model-generated traces, and design three retraining-free interventions that structurally guide reasoning and improve accuracy.
### rebuttal summary
1. To address concerns about model versioning and size, we paired each reasoning model with the closest non-reasoning model from the same family and release period. The observed SR performance gap persisted under these controls.
2. On the use of GPT-4o as a judge, we validated GPT-4o’s judgments against three independent blind annotators on 100 samples, achieving consistent agreement (accuracy = 1.0, κ = 1.0).
3. To verify the annotation reliability, we added inter-annotator agreement statistics for our failure mode classification, reporting a Cohen’s κ of 0.86 across 100 cases.
4. On generalizability of interventions, we applied the combined intervention to 20 AIME 2024 math problems, improving GPT-o3’s original accuracy from 45%→65% (+20 points), showing our intervention's applicability.
5. To show each intervention’s gain, we provided full ablation tables (chess) isolating the effects of each intervention.
We also acknowledge that the original title may oversimplify our findings and will adopt a more precise one. In addition, we have included a limitations section. Examples such as the concrete game illustration in Fig. 1 and additional failure cases in Fig. 1b and Appendix G–I were already present in the original paper. While our notations were already well defined, we have provided a consolidated notation table in the rebuttal for ease of reference.

### post-rebuttal review outcomes
In review, Reviewer 3QQD has given us a positive score. Reviewer EhoN also provided a positive score, was satisfied with our rebuttal and title change, and has maintained that score. Reviewer yG8D’s concerns have been addressed, and they have decided to raise their score. Unfortunately, Reviewer KF7a’s evaluation contains multiple factual inaccuracies and a mischaracterization of our methodology. Of the four listed weaknesses, three are based on incorrect premises, and the remaining point is a general remark unrelated to the paper’s content. The rebuttal comment likewise does not engage with the specific clarifications we provided.

---

### Decision · Program_Chairs · 2025-09-17

**Decision:**

Accept (poster)

**Comment:**

The paper evaluates reasoning and non-reasoning models on certain inductive tasks – identifying secret rules for modified versions of games like chess, black jack etc.. It finds that reasoning models perform worse than non-reasoning counterparts in identifying newly constructed rules, even though they are better on the standard rules. The paper hypothesizes that error accumulation in reasoning models leads to these failures. It identifies three key failure modes and manual inspections of model outputs are used to estimate the failure rates for these. The paper also provides a mathematical formulation for how these can negatively affect thinking models, akin to a mental model. Finally the paper proposes certain interventions that tackle these failure modes, through better prompting, few-shot examples and strict budget forcing. These interventions mitigate the error propagation and lead to better performance on these tasks.

---

The reviewers appreciated the finding that thinking models can be worse on the tasks considered, and the controlled experimental setup that helps identify the failure modes. The mathematical formulation, while very simplified, provides a decent mental model for how thinking models might propagate errors. There were few concerns raised in the discussion period, but the author's response seems to have addressed them.

R1: version mismatch in reasoning and non-reasoning models?, intuitive illustrations for theory, ablations for individual interventions, other tasks — *Addressed by re-evaluating on other versions, presenting ablations as a table and showing the benefit of interventions on AIME*

R2: title is misleading, story is more about “guided reasoning > vanilla reasoning”, no inter-rater agreements for annotators reported — *Addressed by changing title and including inter-rater correlation*

R3: heavily rely on GPT-4o as a judge, CoT baseline, unclear visuals — *Addressed by showing agreement between GPT-4o and human annotators, adding CoT baseline*

R4: clearer presentation of results, more examples for motivation, — *Addressed by pointing to figures in the paper*

---

There are still some typos in the paper and the main figures are very hard to parse (reviewers pointed this out too). Furthermore it is not clear if these issues are specific to inductive reasoning or other tasks too. The newly proposed title addresses this to some extent. Overall the paper makes a positive contribution in understanding limitations of thinking models and provides some mitigation strategies that could inspire more research. The recommendation is to accept the paper, and the authors are encouraged to change the title to the new one.